palaeontology/molecular biology

ZooMS, zooarchaeology, palaeontology, archaeology, Australia

**Authors for correspondence:**
Carli Peters
e-mail: peters@shh.mpg.de
Nicole Boivin
e-mail: boivin@shh.mpg.de

[†]Present address: Department of Archaeology, Max Planck Institute for the Science of Human History, Kahlaische Strasse 10, 07745 Jena, Germany.

# Species identification of Australian marsupials using collagen fingerprinting

Carli Peters[1,†], Kristine K. Richter[2], Tiina Manne[3], Joe Dortch[4], Alistair Paterson[4], Kenny Travouillon[5], Julien Louys[6], Gilbert J. Price[7], Michael Petraglia[1,3,6,8], Alison Crowther[1,3] and Nicole Boivin[1,3,8,9,†]

[1]Department of Archaeology, Max Planck Institute for the Science of Human History, Jena, Germany
[2]Department of Anthropology, Harvard University, Cambridge, MA, USA
[3]School of Social Science, The University of Queensland, Brisbane, Qld 4071, Australia
[4]School of Social Sciences, University of Western Australia, Perth, WA 6009, Australia
[5]Western Australian Museum, Collections and Research, 49 Kew Street, Welshpool, WA 6106, Australia
[6]Australian Research Centre for Human Evolution, Griffith University, Nathan, Qld 4111, Australia
[7]School of Earth and Environmental Sciences, The University of Queensland, Brisbane, Qld 4072, Australia
[8]Department of Anthropology, National Museum of Natural History, Smithsonian Institution, Washington, DC, USA
[9]Department of Anthropology and Archaeology, University of Calgary, Calgary, Canada

CP, 0000-0001-7942-6108; KT, 0000-0003-1734-4742; JL, 0000-0001-7539-0689; GJP, 0000-0001-8406-4594

The study of faunal remains from archaeological sites is often complicated by the presence of large numbers of highly fragmented, morphologically unidentifiable bones. In Australia, this is the combined result of harsh preservation conditions and frequent scavenging by marsupial carnivores. The collagen fingerprinting method known as zooarchaeology by mass spectrometry (ZooMS) offers a means to address these challenges and improve identification rates of fragmented bones. Here, we present novel ZooMS peptide markers for 24 extant marsupial and monotreme species that allow for genus-level distinctions between these species. We demonstrate the utility of these new peptide markers by using them to taxonomically identify bone fragments from a nineteenth-century colonial-era pearlshell fishery at Bandicoot Bay, Barrow Island. The suite of peptide biomarkers presented in this study, which focus on a range of ecologically and culturally important species, have the potential to significantly amplify the zooarchaeological and paleontological record of Australia.

# 1. Introduction

Australia is home to an extremely rich and unique fauna [1], with more than 85% of its terrestrial mammal species classed as endemic [2]. It is the only region globally, other than Papua New Guinea, where marsupials, placentals and monotremes coexist [3]. The unique nature of Australian terrestrial fauna is the outcome of an evolutionary trajectory strongly shaped by the isolation of the Australian continent from Antarctica *ca* 40 Myr ago [3,4]. Among the best-recognized of Australia's fauna are its marsupials, including macropods such as kangaroos and wallabies (members of the suborder Macropodiformes, generally characterized by their long powerful hind legs and feet), as well as other taxa such as koalas and wombats. Australian marsupials inhabit a broad range of ecosystems spanning the continent's arid inland zones, alpine regions, temperate and tropical rainforest, and coastal wetlands [5], and play key ecological roles in many of the ecosystems they inhabit [2]. In the past, marsupials were an important subsistence resource for Aboriginal communities [6–8], while their bones were also used as raw materials for the creation of tools and other artefacts [9–11]. Research on past Australian terrestrial faunas, and particularly marsupials, can provide insight into early human activity on the Australian continent, enable reconstruction of palaeoenvironmental conditions and shifts in biodiversity over time, and help assess the impact of past climate change.

Archaeologists, palaeontologists and other researchers have uncovered an assortment of faunal remains in Australia, dating from the late Pleistocene to the historical period [12–19]. However, the continent's often harsh environmental conditions [20–22], together with other factors like scavenging by marsupial carnivores [23–25], frequently result in a large number of highly fragmented, morphologically unidentifiable bone fragments in archaeological and palaeontological assemblages. Together with a scarceness of reference materials and a tendency toward osteological similarities between species [18,26–28], these factors complicate the study of faunal remains from Australian sites. Zooarchaeology by mass spectrometry (ZooMS) has provided a means to improve taxonomic identifications of fragmented osteological material at sites around the world [29–33] and offers exciting potential to address these challenges in Australian contexts.

ZooMS is a high-throughput, proteomics-based approach that uses differences in the collagen type I (COL1) protein sequence between taxonomic groups to identify faunal remains [34]. COL1 is the most abundant protein in bone, skin, antler and dentine, and these substrates can thus be successfully targeted using ZooMS. In archaeology, ZooMS is increasingly used to identify morphologically unidentifiable bone fragments [29,35], resulting in an improved ability to reconstruct palaeoenvironmental conditions and shifts in biodiversity over time, help assess the impacts of climate change and anthropogenic activities [30,36,37], track the spread of domesticates [32,38–41], identify ancient hominin remains [29,42] and provenance bone tools [31,43,44]. ZooMS is faster and cheaper than ancient DNA (aDNA)-based approaches [34,45,46] and requires less collagen than radiocarbon or stable isotope analyses [47]. While aDNA is often minimally applicable in hot, humid or tropical contexts [48,49], or when studying older assemblages, proteins can preserve over long time periods [50–52] and are more resistant to harsh environments [34].

The prospects for the application of peptide mass fingerprinting on the Australian continent have only been minimally explored. Buckley *et al.* [53] are so far alone in exploring the potential of ZooMS to taxonomically identify Australian marsupials. Although peptide mass fingerprints have been characterized for only eight extant species and the extinct short-faced kangaroo, *Simosthenurus occidentalis*, preliminary findings suggest that ZooMS is an effective method for taxonomically identifying marsupial remains [53]. Here, we build on this research by characterizing collagen peptide markers for a significantly expanded number of extant and recently extinct marsupial and monotreme species to significantly amplify the potential of ZooMS in Australian contexts. We demonstrate the utility of these new markers for ZooMS analyses of Australian faunal assemblages by using them to taxonomically identify fragmented bones from a nineteenth-century colonial pearlshell fishery at Bandicoot Bay, Barrow Island, in Western Australia. The site was selected to enable evaluation of the utility of ZooMS in the continent's arid zone, where organic preservation conditions are often challenging, and because a thorough zooarchaeological study of the site's fauna had already been completed [15,54], enabling a comparison of osteology and ZooMS results.

# 2. Material and methods

## 2.1. Materials

### 2.1.1. Modern reference specimens

Modern bone samples were collected from the Mammalogy collections of Museums Victoria and the Western Australian Museum, the Zooarchaeology Laboratory of the University of Queensland and the

ARCHE Laboratories at Griffith University. Peptide mass fingerprints and collagen sequences were obtained for the short-beaked echidna (*Tachyglossus aculeatus*) and 23 marsupial species: Tasmanian devil (*Sarcophilus harrisii*), thylacine (*Thylacinus cynocephalus*), koala (*Phascolarctos cinereus*), common wombat (*Vombatus ursinus*), hairy-nosed wombat (*Lasiorhinus* sp.), spectacled hare wallaby (*Lagorchestes conspicillatus*), banded hare wallaby (*Lagostrophus fasciatus*), eastern grey kangaroo (*Macropus giganteus*), western grey kangaroo (*Macropus fuliginosus*), red kangaroo (*Osphranter rufus*), common wallaroo (*Osphranter robustus*), Bennett's wallaby (*Notamacropus rufogriseus*), tammar wallaby (*Notamacropus eugenii*), agile wallaby (*Notamacropus agilis*), western brush wallaby (*Notamacropus irma*), Parma wallaby (*Notamacropus parma*), swamp wallaby (*Wallabia bicolor*), northern brown bandicoot (*Isoodon macrourus*), long-nosed bandicoot (*Perameles nasuta*), common brushtail possum (*Trichosurus vulpecula*), ringtail possum (*Pseudocheirus peregrinus*), brush-tailed phascogale (*Phascogale tapoatafa*) and sugar glider (*Petaurus breviceps*). The museum accession numbers for all sampled specimens are listed in electronic supplementary material, table SI.

### 2.1.2. Archaeological specimens

Archaeological specimens were sampled from a late nineteenth-century (1880s/1890s) pearlshell fishery settlement (D24-001) at Bandicoot Bay, Barrow Island, located *ca* 60 km off the northwest coast of Western Australia [15]. The site was surveyed and excavated in 2013 and 2014 as part of the Barrow Island Archaeology Project [15,54–56].

The faunal assemblage consists of 2922 bone fragments, 810 (27.7%) of which were previously identified to the taxonomic class as a part of the zooarchaeological analysis of the site [54]. This rate of morphological identification reflects the harsh taphonomic conditions at the site, which sits on an exposed floodplain subject to summer temperatures approaching 50°C. Bone fragmentation is also considerable at the site; 66% of the identified remains are between 7 and 28 mm in length, and there is a peak in remains between 13 and 16 mm. Although there is evidence of fresh fragmentation, the uniformity of small specimen fragments, along with limited evidence of trampling, is argued by Dooley *et al.* [54] to be the result of weathering at an open-air site.

Zooarchaeological investigations at the Bandicoot Bay site revealed a broad historical exploitation of local resources evidenced by the presence in the assemblage of the golden bandicoot (*I. auratus barrowensis*), brushtail possum (*T. vulpecula*), spectacled hare wallaby (*L. conspicillatus*) and the common wallaroo (*O. robustus isabellinus*). Chelonioidae (sea turtle), microfauna, bird, fish, crab and shark specimens were also identified [54]. Domesticated animals appear to be absent from the bone assemblage [15,54]. For the present study, 134 morphologically unidentifiable bone fragments from the Bandicoot Bay assemblage were sampled for ZooMS analysis.

## 2.2. Collagen extraction

Collagen was extracted from the modern and archaeological bone samples alongside extraction blank controls. For modern specimens, an acid-insoluble approach was used, in which collagen was extracted based upon previously published methods [34,46]. Bone chips of approximately 30 mg were demineralized in 500 µl of 0.6 M hydrochloric acid (HCl) for 48 h. The supernatant was removed, after which the samples were washed three times in 200 µl of 50 mM ammonium bicarbonate (AmBic). Then, the samples were heated at 65°C in 100 µl of 50 mM AmBic. The resulting supernatant was digested with 1 µl of 0.4 µl µg$^{-1}$ trypsin solution (Pierce™ Trypsin Protease, Thermo Scientific) for 18 h at 37°C. Subsequent to enzymatic digestion, peptides were purified and concentrated using C18 ZipTips (Pierce™ C18 Tips, Thermo Scientific).

For archaeological specimens, we employed an acid-soluble approach based on Van der Sluis *et al.* [57]. Bone chips of approximately 30 mg were demineralized in 500 µl of 0.6 M HCl for one week, after which the supernatant was transferred to a 30 kDa ultrafilter (Sartorius, Vivaspin®) and centrifuged until completely passed through the filter. Five hundred microlitres AmBic was then added to the ultrafilter and the samples were centrifuged a second time. The filtrates were resuspended in 100 µl of AmBic followed by digestion and peptide purification as described above. Samples with sufficient collagen preservation for ZooMS were reanalysed with the previously described acid-insoluble approach to get higher quality spectra. The exact protocols are described in detail in Wang *et al.* [47] and are publicly available on protocols.io [58,59].

## 2.3. Matrix-assisted laser desorption/ionization–tandem time of flight mass spectrometry

Modern reference samples were spotted in triplicate onto an MTP AnchorChip 384-target plate, together with matrix solution (10 mg of α-cyano-4-hydroxycinnamic acid in 7 ml of 85% acetonitrile (ACN)/0.1% trifluoracetic acid (TFA)). Archaeological samples were mixed with matrix solution (α-cyano-4-hydroxycinnamic acid of 10 mg ml$^{-1}$ in 50% ACN/0.1% TFA) and spotted onto an MTP Groundsteel 384-target plate. All samples were analysed using an Autoflex Speed LRF matrix-assisted laser desorption/ionization–tandem time of flight mass spectrometer (MALDI-TOF-MS, Bruker Daltonics) with a smartbeam-II laser. A SNAP averaging algorithm was used to obtain monoisotopic masses (C: 4.9384, N: 1.3577, O: 1.4773, S: 0.0417, H: 7.7583).

## 2.4. Liquid chromatography with tandem mass spectrometry

For every modern reference species, one sample with a good MALDI spectrum was selected for further analysis using liquid chromatography with tandem mass spectrometry (LC-MS/MS). Twenty microlitres of the final collagen extract was dried down and sent for LC-MS/MS analysis at the Functional Genomics Center Zurich. LC-MS/MS was conducted using a Q-Exactive HF mass spectrometer (Thermo Scientific) coupled with an ACQUITY UPLC M-Class system (Waters AG). Solvent composition at the two channels was 0.1% formic acid for channel A and 0.1% formic acid, 99.9% ACN for channel B. Column temperature was 50°C. For each sample, 4 μl of peptides was loaded on a commercial MZ Symmetry C18 Trap Column (100 Å, 5 μm, 180 μm × 20 mm, Waters) followed by nanoEase MZ C18 HSS T3 Column (100 Å, 1.8 μm, 75 μm × 250 mm, Waters). The peptides were eluted at a flow rate of 300 nl min$^{-1}$ by a gradient from 5 to 40% B in 120 min and 98% B in 5 min. The column was cleaned after each run with 98% solvent B for 5 min and holding 98% B for 8 min prior to re-establishing loading condition. The mass spectrometers were operated in data-dependent mode performing higher energy collision dissociation (HCD) fragmentation on the 12 most intense signals per cycle. Full-scan MS spectra (300–1500 $m/z$) were acquired at a resolution of 120 000 at 200 $m/z$ after accumulation to a target value (AGC) of 3 000 000, while HCD spectra were acquired at a resolution of 30 000 using a normalized collision energy of 28 (maximum injection time: 50 ms; AGC: 10 000 ions). Unassigned singly charged ions and ions were excluded. Precursor masses previously selected for MS/MS measurement were excluded from further selection for 30 s, and the exclusion window was set at 10 ppm. The samples were acquired using internal lock mass calibration on $m/z$ 371.1012 and 445.1200.

## 2.5. Identification and confirmation of biomarkers

The identification and confirmation of peptide biomarkers were performed following the methodology described in Richter *et al.* [60]. MALDI spectra were visually inspected with FlexAnalysis v. 3.4 (Bruker Daltonics) and compared to a list of published peptide markers for Australian marsupials [53]. When published peptide markers were not available, candidate peptide biomarkers were identified.

Candidate peptide biomarkers were confirmed with LC-MS/MS data analysed in a multi-stage approach using Byonic v. 3.2.0 (Protein Metrics Inc. [61]). First, the product ion spectra were searched against a reference database including the amino acid sequences of COL1A1 and COL1A2 of *P. cinereus* (XP_020853290.1; XP_020855640.1), *V. ursinus* (A0A4X2KF99; A0A4X2M815), *S. harrisii* (G3WK23; G3VSR0) and *Macropus* sp. [62] and common contaminants [63], with the following parameter settings: cleavage sites fully specific C-term R and K; 3 missed cleavages allowed; mass changes: 6 common, 0 rare; common: oxidation on K, M, and P, deamidation on N and Q; no sequence variations allowed; wildcard search disabled. Masses of published and candidate peptide markers were checked to identify the corresponding amino acid sequence (protein FDR 2%, peptide PEP2D score lower than 0.01).

Next, species without confirmed sequence data for all candidate markers were reanalysed using an error-tolerant search strategy to identify novel sequence variants. The following parameter settings were used: cleavage sites fully specific C-term R and K; 2 missed cleavages allowed; mass changes: 3 common, 1 rare; common: oxidation on K, M and P, deamidation on N and Q; rare: all sequence variants allowed; wildcard search disabled. The locations of the peptide markers on the collagen gene were checked and all possible sequence variants and their corresponding masses were recorded (protein FDR 2%, peptide PEP2D score lower than 0.01).

Other proteins in the samples were identified by searching the MS/MS spectral data against the proteomes of *V. ursinus* (UP000314987) and *S. harrisii* (UP000007648) and all sequence data available

in Swissprot, using the following parameter settings: cleavage sites fully specific C-term R and K; 3 missed cleavages allowed; mass changes: 2 common, 1 rare; common: oxidation on K, M and P, deamidation on N and Q; rare: pyro-Glu on N-term E and Q, ammonia-loss on N-term C; no sequence variations allowed; wildcard search disabled; protein FDR 2%. The results were checked for identified bone proteins, other than COL1A1 and COL1A2, and common contaminants.

The results of the first three searches were used to create a new database consisting of (i) the COL1A1 and COL1A2 sequences of the original reference database, (ii) all sequence variants found in the error-tolerant search, (iii) all proteins identified in the whole proteome validation and (iv) common contaminants. The MS/MS data were then analysed with Byonic using this database and the same parameter settings as the first non-error-tolerant search. The protein FDR was set to 2%. Only peptides recurring at least three times, and with a PEP2D score lower than 0.01, were considered confirmed. This resulted in a list of confirmed peptide markers and corresponding peptide sequences.

# 3. Results

All modern reference samples yielded high-quality MALDI and MS/MS spectral data that could be used to characterize collagen peptide markers. Collagen was identified as the main protein component in all samples, and no common contaminants were identified in high quantities (see electronic supplementary material, table II). The ZooMS markers for the studied species are presented in table 1 with the corresponding peptide sequences presented in table 2. Overall, collagen peptide markers generally allow for genus-level distinctions of Australian marsupials, with some limitations within the group of macropods.

## 3.1. Novel ZooMS peptide markers

In addition to the set of peptide markers that is regularly reported for ZooMS studies, we report two additional peptide markers that can be used to distinguish between marsupial taxa. Peptide marker COL1A2 10–42 is represented by $m/z$ 2975 for the majority of the studied reference species. However, this mass value also represents COL1A2 757–789 (G′) in *P. cinereus*, *Lasiorhinus* sp. and *V. ursinus*. Caution is thus needed when interpreting a peak at $m/z$ 2975. It is only possible to confidently assign this peak to either COL1A2 757–789 (G′) or COL1A2 10–42 when another mass peak corresponding to a different COL1A2 757–789 (G′) or COL1A2 10–42 peptide marker has also been identified.

Furthermore, a second novel peptide marker has been identified, peptide marker COL1A2 889–906. This marker is located at the same position in the COL1A2 sequence as the recently reported additional marker to differentiate between bovid taxa [33]. It is thus possible that this novel peptide marker is informative not only for bovids and marsupials, but also for other taxonomic groups.

## 3.2. Marsupial versus monotreme ZooMS markers

Peptide marker COL1A1 508–519 (P1) is often characterized as highly conserved with a peak at $m/z$ 1105 for most terrestrial mammals [65] and at $m/z$ 1079 for cetaceans [45]. For marsupials, this marker is present at $m/z$ 1162, and for *T. aculeatus* at $m/z$ 1120. This difference is particularly interesting as it offers the opportunity to distinguish Australian marsupials and monotremes from other mammalian species on the basis of a single peptide marker. It should be noted, however, that this peptide marker has also been identified at $m/z$ 1162 for many species of birds and reptiles [30], and the reported peptide sequence is identical to the one found in marsupials.

Other peptide markers that have the ability to distinguish between monotremes and marsupials are COL1A2 502–519 (C) and COL1A2 454–483 (E). Both of these markers have been reported at identical mass values in all marsupial species studies ($m/z$ 1598 and $m/z$ 2335, respectively). However, in *T. aculeatus*, these peptide markers were reported at different mass values ($m/z$ 1607 and $m/z$ 2848, respectively). The large offset between the $m/z$ values of peptide marker COL1A2 454–483 (E) for monotremes and marsupials is the result of an amino acid change after a tryptic cut site. In monotremes, the peptide contains a proline following a lysine, resulting in a missed cleavage. The peptide thus contains an additional four amino acids in comparison to the corresponding marsupial marker. This highlights a potential issue with the recently introduced nomenclature system for ZooMS [64], which relies on the location of peptides in the collagen sequence. The change in the location of

**Table 1.** ZooMS markers for Australian marsupial species and the monotreme *T. aculeatus*. Naming of peptide markers follows Brown *et al.* [64]. Masses in italics are not visible in MALDI spectra but are present in LC-MS/MS data. Masses in bold can be used to differentiate monotremes from marsupials.

| | peptide markers | | | | | | | | | | | | | |
|---|---|---|---|---|---|---|---|---|---|---|---|---|---|---|
| | COL1A1 508–519 | COL1A2 978–990 | | COL1A2 484–498 | COL1A2 502–519 | COL1A2 292–309 | COL1A2 793–816 | COL1A2 454–483 | COL1A1 586–618 | | COL1A2 757–789 | | COL1A2 10–42[a] | COL1A2 889–906 |
| | P1 | A | A' | B | C | P2 | D | E | F | F' | G | G' | | |
| *Tachyglossus aculeatus* | **1120** | **1182** | **1198** | 1453 | **1607** | x | **2121** | **2848** | **2873** | **2889** | **2999** | **3015** | **3009** | **1606** |
| *Phascogale tapoatafa* | 1162 | x | x | 1453 | 1598 | **1725** | **2163** | 2335 | 2897 | 2913 | **2929** | **2945** | 2975 | 1652 |
| *Sarcophilus harrisii* | 1162 | **1159** | **1175** | 1453 | 1598 | **1725** | **2177** | 2335 | **2869** | **2885** | **2929** | **2945** | 2975 | 1652 |
| *Thylacinus cynocephalus* | 1162 | **1159** | **1175** | 1453 | *1598* | x | **2121** | 2335 | **2869** | **2885** | **2929** | **2945** | 2975 | 1652 |
| *Lagorchestes conspicillatus* | 1162 | 1150 | 1166 | 1453 | 1598 | x | 2145 | 2335 | 2897 | 2913 | 2943 | 2959 | 2975 | 1652 |
| *Lagostrophus fasciatus* | 1162 | 1150 | 1166 | 1453 | 1598 | 1680 | 2145 | 2335 | **2881** | **2897** | 2943 | 2959 | 2975 | 1652 |
| *Macropus fuliginosus* | 1162 | 1150 | 1166 | 1453 | 1598 | 1680 | 2145 | 2335 | 2897 | 2913 | 2943 | 2959 | **2989** | 1652 |
| *Macropus giganteus* | 1162 | 1150 | 1166 | 1453 | 1598 | 1680 | 2145 | 2335 | 2897 | 2913 | 2943 | 2959 | **2989** | 1652 |
| *Osphranter robustus* | 1162 | 1150 | 1166 | 1453 | 1598 | 1680 | 2145 | 2335 | 2897 | 2913 | 2943 | 2959 | 2975 | 1652 |
| *Osphranter rufus* | *1162* | *1150* | *1166* | *1453* | *1598* | *1680* | *2145* | *2335* | *2897* | *2913* | *2943* | *2959* | *2975* | *1652* |
| *Notamacropus agilis* | 1162 | 1150 | 1166 | 1453 | 1598 | 1680 | 2145 | 2335 | 2897 | 2913 | 2943 | 2959 | 2975 | 1652 |
| *Notamacropus eugenii* | *1162* | *1150* | *1166* | *1453* | *1598* | *1680* | *2145* | *2335* | *2897* | *2913* | *2943* | *2959* | *2959* | **1624** |
| *Notamacropus irma* | *1162* | *1150* | *1166* | **1427** | *1598* | *1680* | *2145* | *2335* | *2897* | *2913* | *2943* | *2959* | *2975* | 1652 |
| *Notamacropus parma* | *1162* | *1150* | *1166* | *1453* | *1598* | *1680* | *2145* | *2335* | *2897* | *2913* | *2943* | *2959* | *2975* | 1652 |
| *Notamacropus rufogriseus* | *1162* | *1150* | *1166* | *1453* | *1598* | *1680* | *2145* | *2335* | *2897* | *2913* | *2943* | *2959* | *2975* | 1652 |
| *Wallabia bicolor* | 1162 | 1150 | 1166 | 1453 | 1598 | 1680 | 2145 | 2335 | 2897 | 2913 | 2943 | 2959 | 2975 | **1624** |
| *Petaurus breviceps* | 1162 | **1159** | **1175** | **1411** | 1598 | **1650** | 2145 | 2335 | 2897 | 2913 | **2971** | **2987** | 2975 | **1624** |
| *Trichosurus vulpecula* | 1162 | **1137** | **1153** | 1453 | 1598 | x | 2145 | 2335 | 2897 | 2913 | **2945** | **2961** | 2975 | **1624** |

(Continued.)

**Table 1.** (*Continued.*)

| | peptide markers | | | | | | | | | | | | | |
|---|---|---|---|---|---|---|---|---|---|---|---|---|---|---|
| | COL1A1 508–519 | COL1A2 978–990 | | COL1A2 484–498 | COL1A2 502–519 | COL1A2 292–309 | COL1A2 793–816 | COL1A2 454–483 | COL1A1 586–618 | | COL1A2 757–789 | | COL1A2 10–42[a] | COL1A2 889–906 |
| | P1 | A | A' | B | C | P2 | D | E | F | F' | G | G' | | |
| *Phascolarctos cinereus* | 1162 | **1159** | **1175** | **1397** | 1598 | **1692** | 2145 | 2335 | **2869** | **2885** | **2959** | **2975** | 2975 | **1624** |
| *Pseudocheirus peregrinus* | 1162 | *1137* | *1153* | **1439** | 1598 | *1650* | **2161** | 2335 | 2897 | 2913 | **2929** | *2945* | 2975 | **1624** |
| *Lasiorhinus* sp. | 1162 | x | x | **1397** | 1598 | **1692** | 2145 | 2335 | **2869** | **2885** | **2959** | **2975** | 2975 | **1624** |
| *Vombatus ursinus* | 1162 | **1159** | **1175** | **1439** | 1598 | *1650* | **2119** | 2335 | 2897 | 2913 | **2959** | **2975** | **2991** | **1624** |
| *Isoodon macrourus* | 1162 | 1150 | 1166 | 1453 | 1598 | x | **2177** | 2335 | 2897 | 2913 | **2957** | **2973** | 2975 | **1624** |
| *Perameles nasuta* | 1162 | *1159* | *1175* | 1453 | 1598 | x | **2177** | 2335 | **2869** | **2885** | **2957** | **2973** | 2975 | **1624** |

[a]Additional marsupial peptide marker first proposed by Buckley *et al.* [53] with tryptic name 2T3.

**Table 2.** Peptide sequences corresponding to ZooMS markers presented in table 1. Naming of peptide markers follows Brown *et al.* [64]. Masses in parentheses represent the mass of the peptide with an additional oxidation. Differences between sequences are in bold and underlined.

| marker | | sequence | mass |
|---|---|---|---|
| COL1A1 508–519 | P1 | GVQGP**A**GPQGPR | 1120 |
| | | GVQGP**P**GPQGPR | 1162 |
| COL1A2 978–990 | A | **PGNA**GA**V**GPAGL**R** | 1137 (1153) |
| | | **PGQA**GA**V**GPAGL**R** | 1150 (1166) |
| | | **PGHA**GA**V**GPAGL**R** | 1159 (1175) |
| | | **SGQPG**T**V**GPAGV**R** | 1182 (1198) |
| COL1A2 484–498 | B | G**VA**GEFG**LP**GPAGPR | 1397 |
| | | G**PA**GEFG**LP**GPAGPR | 1411 |
| | | G**SP**GEFG**LP**GPAGPR | 1427 |
| | | G**VP**GEFG**LP**GPAGPR | 1439 |
| | | G**LP**GEFG**LP**GPAGPR | 1453 |
| COL1A2 502–519 | C | GPPGESGA**V**GPTG**S**IG**S**R | 1598 |
| | | GPPGESGA**A**GPTG**P**LG**N**R | 1607 |
| COL1A2 292–309 | P2 | GPNGEPGSTGP**S**GPPGLR | 1650 |
| | | GPNGEPGSTGP**T**GPPGLR | 1680 |
| | | GPNGEPGSTGP**P**GPPGLR | 1692 |
| | | GPNGEPGSTGP**M**GPPGLR | 1725 |
| COL1A2 793–816 | D | GLPGVSG**SL**GEPGPLGI**A**GP**A**GAR | 2119 |
| | | GLPGVSG**SV**GEPGPLGI**A**GP**A**GAR | 2121 |
| | | GLPGVSG**GL**GEPGPLGL**S**GP**S**GAR | 2121 |
| | | GLPGVSG**AL**GEPGPLGI**A**GP**P**GAR | 2145 |
| | | GLPGVSG**SL**GEPGPLGI**A**GP**P**GAR | 2161 |
| | | GLPGVSG**SV**GEPGPLGI**S**GP**P**GAR | 2163 |
| | | GLPGVSG**SL**GEPGPLGI**S**GP**P**GAR | 2177 |
| COL1A2 454–483 | E | GEQGPAGPPGFQGLPGPSGPAGE**G**GK | 2335 |
| | | GEQGPAGPPGFQGLPGPSGPAGE**V**GKPGER | 2848 |
| COL1A1 586–618 | F | GLTGPIGPPGPAG**PS**GDKGESGPSGP**A**GPTGAR | 2869 (2885) |
| | | GLTGPIGPPGPAG**TS**GDKGESGPSGP**A**GPTGAR | 2873 (2889) |
| | | GLTGPIGPPGPAG**PA**GDKGESGPSGP**V**GPTGAR | 2881 (2897) |
| | | GLTGPIGPPGPAG**PS**GDKGESGPSGP**V**GPTGAR | 2897 (2913) |
| COL1A2 757–789 | G | GP**P**GE**A**GA**S**GPPGSSGPQGL**L**GAPGILGLPGSR | 2929 (2945) |
| | | GP**P**GE**A**GA**T**GPPGSSGPQGL**L**GAPGILGLPGSR | 2943 (2959) |
| | | GP**E**GE**A**GA**S**GPPGSSGPQGL**L**GAPGILGLPGSR | 2945 (2961) |
| | | GP**P**GE**S**GA**V**GPPGSSGPQGL**L**GAPGILGLPGSR | 2957 (2973) |
| | | GP**P**GE**S**GA**T**GPPGSSGPQGL**L**GAPGILGLPGSR | 2959 (2975) |
| | | GP**P**GE**S**GA**L**GPPGSSGPQGL**L**GAPGILGLPGSR | 2971 (2987) |
| | | GP**P**GE**A**GA**T**GPPGSSGPQGL**W**GAPGILGLPGSR | 2999 (3015) |
| COL1A2 10–42 | | GPPGA**S**GPPGAQGFQGPAGEPGE**P**GQTGPAGA**R** | 2975 |
| | | GPPGA**T**GPPGAQGFQGPAGEPGE**P**GQTGPAGA**R** | 2989 |
| | | GPPGA**S**GPPGAQGFQGPAGEPGE**P**GQTGPAG**S**R | 2991 |
| | | GPPGA**S**GPPGAQGFQGPAGEPGE**D**GQTGPAGA**R** | 3009 |
| COL1A2 889–906 | | GEPGP**V**GSVGPVGP**T**GAR | 1606 |
| | | GEPGP**A**GSVGPVGP**F**GAR | 1624 |
| | | GEPGP**V**GSVGPVGP**F**GAR | 1652 |

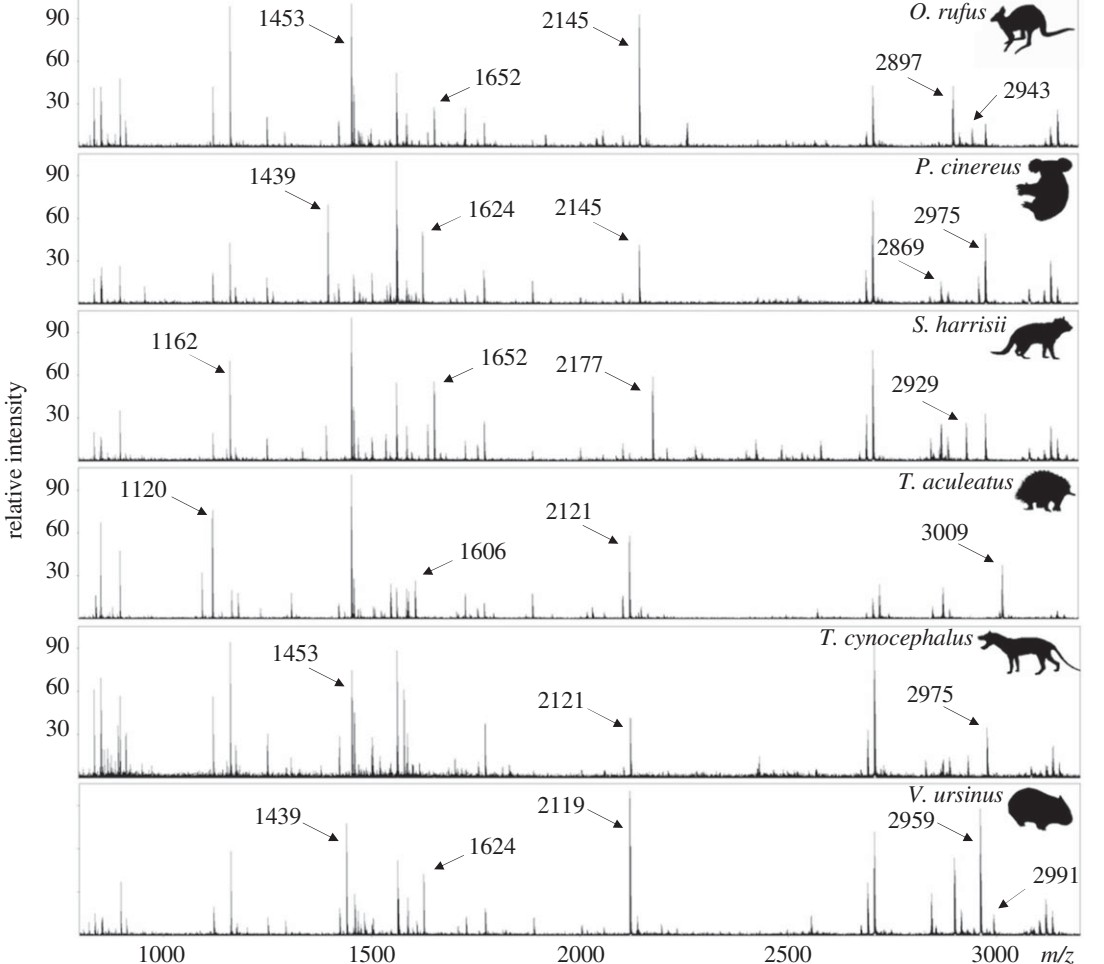

**Figure 1.** Examples of peptide mass fingerprints for *O. rufus*, *P. cinereus*, *S. harrisii*, *T. aculeatus*, *T. cynocephalus* and *V. ursinus*.

the tryptic cut site, as shown above, also results in a change of the starting location of the next peptide in the collagen sequence.

Next to the differences at COL1A1 508–519 (P1), COL1A2 502–519 (C) and COL1A2 454–483 (E), the monotreme *T. aculeatus* also differs from the studied marsupials in the majority of the other peptide markers. The observed differences between *T. aculeatus* and the other species studied are most likely the result of the unique evolutionary trajectory of *T. aculeatus* [4], which is the only monotreme species included in this study. However, caution should be taken when extrapolating these results to Ornithorhynchidae, the other monotreme family, as it is possible that these peptide markers may differ between taxa.

## 3.3. Marsupial ZooMS markers

Peptide markers that show a high level of variation between marsupial taxa are COL1A2 978–990 (A), COL1A2 484–498 (B), COL1A2 292–309 (P2), COL1A2 793–816 (D), COL1A1 586–618 (F/F′), COL1A2 757–789 (G/G′), COL1A2 10–42 and COL1A2 889–906. With the exception of *I. macrourus*, peptide marker COL1A2 978–990 (A) has the ability to distinguish macropods from other marsupials. A combination of the other peptide markers can be used to differentiate between other marsupial species (table 1 and figure 1).

Peptide marker COL1A2 (P2) shows a high level of variation between marsupial species. However, for most species, this marker was not visible in the MALDI spectra and was only identified in the MS/MS spectral data in low quantities. Therefore, this marker has only been reported for species where the peptide sequence could be confirmed in the final Byonic search. Peptide marker COL1A2 793–816 (D) also has the ability to differentiate between marsupial species. It must be noted, however, that the peaks at *m/z* 2161 and *m/z* 2177 are not mutually exclusive. For these peptide sequences,

**Table 3.** Peptide marker differences between macropods. Peptide markers in bold represent those that can be used to distinguish between taxa.

| | COL1A2 484–498 | COL1A1 586–618 | COL1A2 10–42 | COL1A2 889–906 |
|---|---|---|---|---|
| *L. conspicillatus* | 1453 | 2897/2913 | 2975 | 1652 |
| *L. fasciatus* | 1453 | **2881/2897** | 2975 | 1652 |
| *Macropus* sp. | 1453 | 2897/2913 | **2989** | 1652 |
| *Osphranter* sp. | 1453 | 2897/2913 | 2975 | 1652 |
| *Notamacropus* sp.[a] | 1453 | 2897/2913 | 2975 | 1652 |
| *N. irma* | **1427** | 2897/2913 | 2975 | 1652 |
| *N. eugenii* | 1453 | 2897/2913 | 2975 | **1624** |
| *W. bicolor* | 1453 | 2897/2913 | 2975 | **1624** |

[a]All species of the genus except for *N. irma* and *N. eugenii*.

peptides with both two and three oxidations of proline are identified in the MS/MS data. A peak at $m/z$ 2177 could, in theory, also match to the peptide sequence reported for $m/z$ 2161, and vice versa. Furthermore, the peptide sequence for *V. ursinus* is visible in the MS/MS data with both two and three oxidations. For most species, the variant with three oxidations is most visible in the MALDI spectra, but for *V. ursinus*, the variant with two oxidations was most visible. We therefore reported this marker at $m/z$ 2119, corresponding to the variant with two oxidations.

Peptide marker COL1A2 757–789 (G/G′) is highly diverse between marsupial species and thus holds the potential to be used to uniquely identify taxonomic groups. However, some of the masses are identical for different peptide sequences. The peak at $m/z$ 2945 can represent both COL1A2 757–789 (G′) for *P. tapoatafa*, *S. harrisii*, *T. cynocephalys* and *P. peregrinus*, as well as COL1A2 757–789 (G) for *T. vulpecula*. Similarly, $m/z$ 2959 can represent COL1A2 757–789 (G′) for macropods as well as COL1A2 757–789 (G) for *P. cinereus*, *Lasiorhinus* sp. and *V. ursinus*. These $m/z$ values can thus only be reliably used to make taxonomic identifications when found in combination with their corresponding G or G′ masses.

## 3.4. Using ZooMS to identify macropods

The ZooMS spectra of all studied macropods (*L. conspicillatus*, *L. fasciatus*, *M. giganteus*, *M. fuliginosus*, *O. rufus*, *O. robustus*, *N. rufogriseus*, *N. eugenii*, *N. agilis*, *N. irma*, *N. parma* and *W. bicolor*) are largely similar, with the majority of the studied species characterized by identical peptide markers. However, it is possible to identify some species based on differences in their peptide markers (table 3). Peptide marker COL1A2 484–498 (B) at $m/z$ 1427 (sequence: GSPGEFGKOGPAGPR) can be used to identify *N. irma*. This peptide marker is located at $m/z$ 1453 (sequence: GLPGEFGLPGPAGPR) in other macropods (figure 2). *N. eugenii* and *W. bicolor* can be distinguished from other macropods on the basis of peptide marker COL1A2 292–309 (P2), with the former two at $m/z$ 1625, compared to $m/z$ 1652 in other macropods. Furthermore, a difference at peptide marker COL1A1 586–618 (F/F′) has been observed between *L. fasciatus* ($m/z$ 2881/2897) and other macropods ($m/z$ 2897/2913). However, because of the partial overlap between these $m/z$ values, it is only possible to confidently assign this peptide marker when both peaks are visible in the MALDI spectra. Finally, $m/z$ 2989, attributed to COL1A2 10–42, can be used to separate *Macropus* (*M. giganteus* and *M. fuliginosus*) from other macropods in which this peptide marker corresponds to $m/z$ 2975.

LC-MS/MS analysis provides a further opportunity to differentiate between macropod species with identical ZooMS peptide markers. Differences have been observed between the amino acid sequence of peptide COL1A2 671–700 for *L. conspicillatus* (sequence GENG**A**VGPTGPVGAAGP**S**GPNGPPGPVGGR) and all other macropod species (sequence GENG**V**VGPTGPVGAAGP**A**GPNGPPGPVGGR). However, the corresponding $m/z$ peaks were not detectable in the MALDI spectra. *L. conspicillatus* is thus only identifiable on the basis of peptide sequence data obtained through LC-MS/MS analysis.

## 3.5. Collagen fingerprinting of archaeological specimens

Of the 134 archaeological samples analysed from Bandicoot Bay, 43 samples (32%) had sufficient collagen preservation to make taxonomic identifications using ZooMS (table 4). Macropods ($n = 36$) make up the

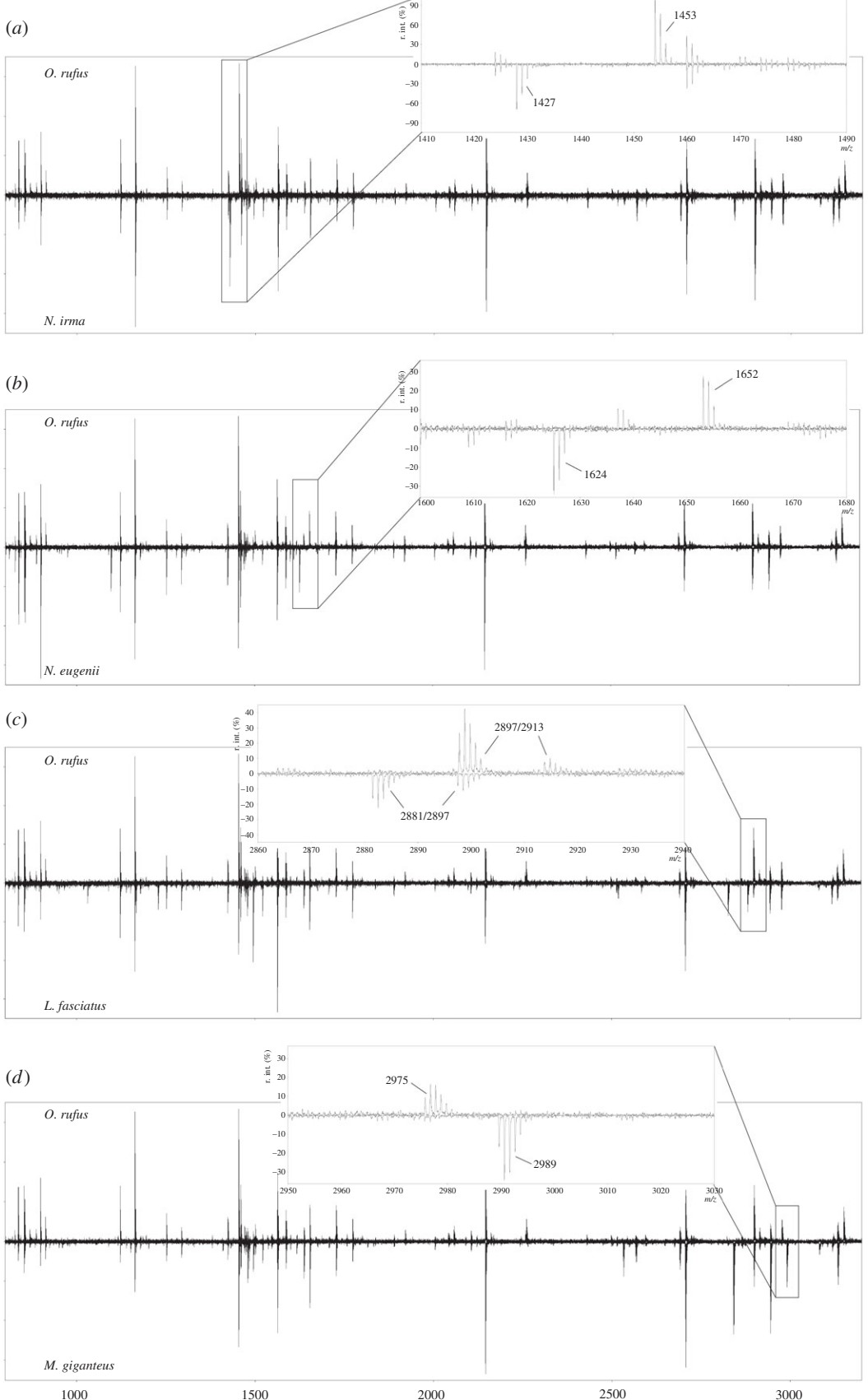

**Figure 2.** Example of differences in peptide mass fingerprints between macropods: (*a*) the difference at COL1A2 484–498 (B) between *O. rufus* and *N. irma*; (*b*) the difference at COL1A2 292–309 (P2) between *O. rufus* and *W. bicolor*; (*c*) the difference at COL1A1 586–618 (F/F′) between *O. rufus* and *L. fasciatus*; (*d*) the difference at COL1A2 10–42 between *O. rufus* and *M. giganteus*.

**Table 4.** ZooMS identifications of analysed archaeological specimens. Morphological ID refers to taxa identified in the Bandicoot Bay zooarchaeological assemblage [15,54]. Samples that failed or were unidentifiable with ZooMS reference data were excluded ($n = 85$).

| ZooMS ID | ZooMS NISP | morphological ID |
|---|---|---|
| *Isoodon* sp. | 3 | *Isoodon auratus barrowensis* |
| *Trichosurus vulpecula* | 1 | *Trichosurus vulpecula* |
| Macropodidae | 36 | *Lagorchestes conspicillatus* |
| | | *Osphranter robustus isabellinus* |
| Bovidae/Cervidae | 1 | |
| *Chelonia mydas* | 2 | Chelonioidea |
| total | 43 | |

bulk of the analysed bone fragments. We can exclude *L. fasciatus*, *M. giganteus*, *M. fuliginosus*, *N. irma*, *N. eugenii* and *W. bicolor* as the origin of these samples on the basis of peptide markers COL1A2 484–498 (B), COL1A1 586–618 (F), COL1A2 10–42 and COL1A2 889–906, which leaves *L. conspicillatus*, *O. robustus*, *O. rufus*, *N. agilis*, *N. parma* and *N. rufogriseus* as possible origins for these bone fragments. It is most likely that the identified macropods represent the two species already identified in the zooarchaeological assemblage, *L. conspicillatus* (spectacled hare wallaby) and *O. robustus isabellinus* (common wallaroo). Three specimens are assigned to *Isoodon* sp.; on the basis of the zooarchaeological identifications of *I. auratus barrowensis* (golden bandicoot), the ZooMS identified specimens most likely represent this species. The collection of a reference sample of *I. auratus barrowensis* would be advantageous in this case. However, this was outside of the scope of the current study, for which this case study served solely as a test of the developed peptide markers. One specimen has been identified as *T. vulpecula* (brushtail possum).

Using previously published peptide markers [66], two specimens were identified as *Chelonia mydas* (green sea turtle). While the presence of sea turtles at Bandicoot Bay was known from zooarchaeological investigations [15,54], the exact species was unknown. In addition, one Bandicoot Bay specimen was identified as Bovidae/Cervidae, fauna that are not present in the wild on Barrow Island. The specimen is most likely a bovid, as only limited introduced cervid populations have been established in Australia, and none in north-western Australia. Possible bovid species that this specimen could represent are sheep and goat, which are known to have been present during the colonial period as livestock, or perhaps cattle, but only if it was brought in as processed, barrelled meat. Since domesticated species were not identified at the site through standard zooarchaeological methods [54], this specimen represents the first reported domesticated animal at Bandicoot Bay and provides evidence for food provisioning.

In addition to the 43 samples that were taxonomically identifiable using ZooMS, six specimens had spectra that were not attributable to any known taxa. Based on the lack of the characteristic COL1A1 508–519 (P1) marker, these specimens are tentatively identified as fish or bird specimens. Furthermore, seven samples only showed a peak at peptide marker COL1A1 508–519 (P1) at $m/z$ 1162, but lacked other peptide markers to make taxonomic identifications. Overall, despite analysing only 4.6% of the total faunal fragments, two previously unidentified taxa (*C. mydas*, and a probable bovid) were identified using ZooMS, increasing the total number of identified taxa at the site from four to six.

## 4. Discussion

The novel set of reference peptide markers developed for this study allow for genus-level identifications of Australian marsupials using ZooMS, with some limitations only with regards to macropods. Collagen fingerprinting of fragmented bones from Bandicoot Bay, Barrow Island has shown the utility of these peptide markers, and we were able to successfully identify bone fragments that were not able to be morphologically identified, highlighting the potential of ZooMS in Australian contexts even in extremely harsh environments. The large-scale application of ZooMS, combined with zooarchaeology, has tremendous potential to contribute to the study of biodiversity trends, past subsistence strategies and material culture at Australian sites.

## 4.1. ZooMS insights at Bandicoot Bay

Despite the poor preservation of the faunal material at Bandicoot Bay, ZooMS was successfully employed to identify a range of marsupials, all of which were previously reported based on standard zooarchaeological techniques. This is an important finding, highlighting the potential of ZooMS in Australia even under harsh taphonomic conditions. The zooarchaeological record at the site was also further expanded with the identification of two green turtle specimens, as well as the first reported domesticated species from the site. While not identified at the site using morphological techniques, the presence of domesticated bovid makes sense in light of the kinds of foods imported to pearling stations, such as salted beef, pork, mutton and other barrelled meats [67–69]. The identification of new species in the Bandicoot Bay assemblage through ZooMS highlights the added value of collagen fingerprinting, especially when applied alongside more standard zooarchaeological investigations using osteological methods. Only a small number of archaeological specimens from Bandicoot Bay were analysed using ZooMS; a much larger study would likely yield further identifications not made on the basis of standard zooarchaeological techniques.

The identification of *C. mydas* (green sea turtle) provides valuable insight into seasonal site use at Bandicoot Bay. Six of the world's seven turtle species visit Barrow Island, with three of these species (*C. mydas*, *Natator depressus* (flatback sea turtle), *Eretmochelys imbricata* (hawksbill sea turtle)) found annually nesting along Barrow Island's sandy beaches [70]. Although *C. mydas* has been observed feeding along the west coast of Barrow Island throughout the year, its population numbers peak between November and March during the nesting season [71]. Nesting primarily takes place along the east coast of Barrow Island and in the sheltered beaches within Little Bandicoot Bay [71], adjacent to the Bandicoot Bay archaeological site. Turtles may have been targeted in open water but the proximity of the archaeological site to a known *C. mydas* nesting location, along with ZooMS identifications of *C. mydas*, suggests the site was occupied during the summer pearling season. This timing supports Dooley *et al.*'s [54] suggestion that the site may represent mid-season provisioning of a pearling lugger and reveals how Aboriginal labourers targeted turtle, which is a highly valued food. This further adds to the story of survival for indentured Aboriginal divers on Barrow Island.

## 4.2. Comparison to published markers

The results from this study broadly align with the incomplete marker profiles published previously for Australian marsupials [53]. Particularly interesting is peptide marker COL1A1 586–618 (F/F′), which differs between *L. fasciatus* at *m/z* 2881/2897 and other macropods at *m/z* 2897/2913. Incomplete published marker profiles for *S. occidentalis*, the extinct short-faced kangaroo, also presented the peaks for COL1A1 586–618 (F/F′) at *m/z* 2881/2897 [53], resembling *L. fasciatus*. Both morphological [72–74] and genetic studies [75–78] have proposed *L. fasciatus* as the closest living relative of the extinct sthenurine kangaroos, and the collagen peptide sequences identified here agree with this proposed close phylogenetic relationship.

There are five instances in which our data do not match the incomplete marker profiles published by Buckley *et al.* [53]. In two cases, the differences stem from increasing the number of reference samples and therefore providing increased taxonomic resolution. Our analysis of collagen sequence data and LC-MS/MS data confirm that there are differences in peptide marker COL1A2 10–42 between *Macropus* and *Osphranter*, contradicting previous uniform reporting of this peptide marker for these genera [53]. We also analysed MALDI spectra and LC-MS/MS data from an additional species in the genus *Isoodon* and found that it is possible to distinguish between both on the basis of peptide markers COL1A2 793–816 (D) and COL1A1 586–618 (F/F′).

In three other cases, our markers directly disagreed with previously published markers. We found different *m/z* values for peptide marker COL1A2 10–42 for *P. peregrinus* reported at *m/z* 2987 by Buckley *et al.*, and for peptide markers COL1A2 793–816 (D), COL1A2 757–789 (G/G′) and COL1A2 10–42 for *V. ursinus*, reported at *m/z* 2161, 2929, and 2959, respectively [53]. Our analysis of collagen sequence data and LC-MS/MS data confirmed the peptide markers we proposed on the basis of the MALDI-TOF-MS spectral data and aligns with genetic data available for *V. ursinus*. In the case of *P. peregrinus*, our proposed peptide marker aligns with the other species reported in this study, and no other species have peptide marker COL1A2 10–42 at *m/z* 2987. We therefore argue that the peptide markers presented in this study, that differ from the ones reported previously, are correct. Since neither the MALDI spectra nor the raw MS/MS data were openly available for the previous study, we are unable to re-evaluate the previously published markers. We thus stress the importance of

confirming peptide biomarkers using collagen sequence data obtained through LC-MS/MS analysis, and where possible genetic data, as well as making MALDI spectra and raw MS/MS data openly available.

## 4.3. Challenges and future prospects

One of the main challenges for the identification of marsupials using ZooMS is that some species are only distinguishable on the basis of a single peptide marker. This often requires high-quality spectral data, in which the high mass peaks are clearly defined. However, collagen preservation has proven to be problematic in Australian faunal assemblages [20,21], and our own results support this finding. Collagen preservation in the Bandicoot Bay assemblage was variable; in some specimens, collagen was reasonably well preserved and spectra were suitable to make taxonomic identifications, while poorer preservation in other specimens meant that resolution of spectra was not sufficient for ZooMS identifications. These issues surrounding collagen preservation can be expected to increase considerably when older assemblages are analysed. The recent development of pre-screening techniques, such as Fourier transform infrared spectroscopy, that can be used to assess the molecular preservation of fossil material [79–82] could offer a partial solution to this challenge. These methods allow for the rapid identification of well-preserved specimens that are suitable for proteomic analysis.

As with other ZooMS markers that have been developed, the markers presented here do not show a straightforward relationship between phylogenetic distance and sequence variation. COL1 is functionally constrained and therefore exhibits a slow rate of evolutionary change and accumulation of sequence mutations. Accordingly, only a small percentage of the COL1 sequence produces usable ZooMS markers, with the result that divergence time alone is not a sufficient predictor of the ability to discriminate taxa using ZooMS. Within marsupials, genus-level identifications are possible within some families. The genera *Phascogale* and *Sarcophilus* (diverged 13–21.2 Ma [83]) from the family Dasyuridae can be distinguished, for example, as can the genera *Isoodon* and *Perameles* (diverged 12.1–4.8 Ma [83]) from the family Peramelidae. In the Macropodidae family, the genus *Lagostraphus* can be distinguished from the rest of the genera in the family (diverged 25.7–11.9 Ma [75]). However, for *Macropus*, *Notamacropus*, *Osphranter* and *Wallabia* (common ancestor 13.7–8 Ma [75]), divergence time does not line up with the number of variations in the COL1 sequence. For example, within *Notamacropus*, *N. eugenii* and *N. agilis* (diverged less than 2 Ma [75]) can be distinguished, while *N. eugenii* and *W. bicolor*, which are more distantly related, have identical ZooMS marker profiles.

Although it is not possible to uniquely identify every marsupial species using ZooMS alone, peptide mass fingerprinting combined with biogeographical and zooarchaeological data does provide the opportunity to significantly increase the accuracy of taxonomic identifications. For example, while eastern and western grey kangaroos (*M. giganteus* and *M. fuliginosus*, respectively) share an identical set of ZooMS peptide markers, the eastern species is today found in the eastern states of Australia, while the western species occurs in the southern and western parts of Australia [84]. However, the potential for recent local extinctions as well as range shifts should not be overlooked [19,85–87]. Eastern and western kangaroo ranges overlap in south-central Australia, and a recent aDNA study has demonstrated that, historically, both species were found on Kangaroo Island, whereas it was previously thought that only the western variety had roamed there [88]. Nonetheless, species biogeography can in some cases still help support the attribution of more specific taxonomic identifications, if used cautiously and bearing the clear caveats in mind.

The development of ZooMS peptide markers for Australian marsupials is significant for zooarchaeological endeavours on the continent. Marsupial taxa form a significant component of Australian zooarchaeological and palaeontological assemblages, yet due to the challenges of discriminating between related taxa, they are currently rarely identified beyond the very broad family level [18,28,87]. This significantly limits the interpretive power of zooarchaeological and palaeontological studies on the continent [28]. The current set of peptide markers holds promise for enhancing research into marsupial biodiversity change in Australia, as well as palaeoenvironmental change and human behaviour.

To further expand the capabilities of ZooMS in Australian contexts, it will be necessary to develop peptide markers for other Australian taxa, such as other extant monotremes. The highly fragmented nature of bone assemblages at Australian sites due to poor preservation conditions, carnivore activity and extensive carcass processing by humans necessitated by the limited food resources available in arid environments [89] will likely make ZooMS an important tool in the repertoire available to Australian zooarchaeologists and palaeontologists moving forward. ZooMS will also be useful when morphological similarities make it challenging to differentiate between genera. Macropods, for

example, can be represented by multiple extant genera in a single region [7,90]. These are challenging to distinguish with morphological approaches unless dental remains are present and even then, heavy toothwear may confound identifications [28,91]. This inherently limits the identification of faunal material through morphological approaches, leaving a significant proportion of postcranial bones from Australian archaeological sites unidentified beyond family level. By offering the ability to undertake higher level taxonomic attributions of a broader range of faunal remains, ZooMS can help to significantly enhance Australian faunal datasets.

Future extension of Australian ZooMS markers to include small mammal species will be particularly useful. ZooMS has the potential to address a critical deficit in examining biodiversity trends in native rodents, bandicoots and dasyurids on the continent. These are often overlooked in both archaeological and palaeontological sites due to difficulties in identification. Nevertheless, they are critical environmental indicators [92,93] and were an important food source for some Australian indigenous groups [94–96]. Improved identification rates of these species in early contact sites is also important for tracking the spread of non-native species in Australia, information that is also critical for modern conservation efforts [97]. Furthermore, the possibility of identifying extinct megafauna species using collagen fingerprints holds significant potential to aid the study of megafauna extinctions on the continent. One of the main issues for understanding these extinctions in Australia is the significant knowledge gaps that exist for many species. Data on species biochronology and palaeobiogeography are mostly patchy, with many species only having been reported at a handful of sites [18,98,99]. ZooMS offers a significant opportunity to address these issues by improving identification rates of fragmented bones.

The peptide biomarkers developed for this study hold tremendous potential for the increasing application of ZooMS in Australia, where the method has so far been minimally applied relative to European contexts. Future years will see the important set of reference markers published here added to and enhanced, offering significant potential for addressing new research questions and themes, and enhancing the ability of researchers to examine topics of current interest in Australian archaeology and palaeontology.

Ethics. The handling, sampling and analysis of modern and archaeological specimens were conducted in accordance with CITES regulations. No live animals were included in the study.

Data accessibility. MALDI-TOF-MS spectra for modern references (https://doi.org/10.5281/zenodo.5040049) [100] and archaeological specimens (https://doi.org/10.5281/zenodo.5040055) [101] have been uploaded to Zenodo and are publicly available. MS/MS data files are available at PXD027107 and were uploaded through MassIVE (MSV000087755, https://doi.org/10.25356/C5TC2H) [102]. The URL link to the dataset is ftp://massive.ucsd.edu/MSV000087755/.

Authors' contributions. C.P., K.K.R., T.M., J.D., J.L., G.J.P., M.P., A.C. and N.B. conceived and designed the study; C.P. performed ZooMS analysis and prepared samples for LC-MS/MS analysis; C.P. and K.K.R. analysed ZooMS and LC-MS/MS data; T.M., A.P., K.T., J.L. and G.J.P. provided samples; C.P. and N.B. wrote the paper, with critical input from all authors. All authors reviewed and approved the manuscript.

Competing interests. We declare we have no competing interests.

Funding. Open access funding provided by the Max Planck Society.

This research was supported by the Australian Research Council (grant no. DP130100802, awarded to T.M. and A.P.; grant no. DE150101597, awarded to T.M.; and grant no. FT150100168, awarded to A.P.).

Acknowledgements. We are grateful to the Mammalogy collection of the Western Australian Museum, and Kylea Clarke from the Mammalogy collection of Museums Victoria for granting us access to their collections. We would also like to thank Sandra Hebestreit for the technical assistance with ZooMS extractions. Finally, we would like to thank the two anonymous reviewers for their comments and suggestions, which helped improve the paper.

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
