## [Peer Review File · Royal Society Open Science]

Review History

RSOS-211229.R0 (Original submission)

Review form: Reviewer 1

Is the manuscript scientifically sound in its present form?

Yes

Are the interpretations and conclusions justified by the results?

Yes

Is the language acceptable?

Yes

Do you have any ethical concerns with this paper?

No

Have you any concerns about statistical analyses in this paper?

No

Recommendation?

Accept with minor revision (please list in comments)

Comments to the Author(s)

Review of Peters et al., "Species Identification of Australian Marsupials using Collagen Fingerprinting"

Peters et al. present an extremely thorough and nicely written manuscript that sets out a suite of collagen type 1 biomarkers for 24 marsupials and a monotreme from Australia, the vast majority of biomarkers of which are new to the field. They then apply these biomarkers to a set of archaeological samples from Barrow Island (WA), from a 19th century pearlshell factory, in order to attempt collagen-derived taxonomic identification. The article vastly improves knowledge of marsupial (and monotreme) collagen biomarkers, as well as correcting some previously incorrectly assigned biomarkers from earlier in the 'evolution' of the technique. Overall, this is an extremely useful study that should be published immediately to give the greatest benefits to the zooarchaeological community. May I also thank the authors for depositing the raw MALDI and MS/MS data in an open access folder. I have no major issues with the manuscript, which was an enjoyable read, and just a few minor comments that I hope will strengthen the paper and be relatively easy for the authors to address.

Minor points

- Abstract: "We show that the application of ZooMS has the potential to significantly amplify the zooarchaeological and paleontological record of Australia" – I don't agree that your results for the archaeological assemblage in this study shows the potential to significantly amplify the zooarchaeological and paleontological record of Australia. I believe your biomarkers can, but your application of them in this case does not 'significantly amplify' the records of Australia. Your samples, with support of prior morphological analysis, increased the number of taxa known on the site from 4 to 6. This is not a large enough sample size to claim such significance. Please adapt this sentence to show the real strengths of the paper, which is the huge suite of biomarkers, rather than the specific application to the Barrow Island samples
- It would be useful for the readers if you were able to very briefly define 'macropods' in the Introduction
- Page 3, line 2: suggestion to include anthropogenic activities. I.e. "...enable reconstruction of palaeoenvironmental conditions and shifts in biodiversity over time, and help assess the impact of past climate change and anthropogenic activities."
- Page 3, line 34: You mention here that you are building on extant species, but you include extinct also, including thylacine. Would it be better to state "extant and extinct"?
- Page 4, lines 23 onwards: it is striking here that one of the species identified from the morphological assessments of the bone assemblage were not selected for the collagen reference collection. It would be most advantageous to acquire a modern bone sample of golden bandicoot (*I. auratus barrowensis*), but if this is outside of the scope of the current study (which I assume it is) then text should be included somewhere that explicitly notes and explains this omission. For example, in section 3.5.
- Methods: Can you include more precise information about the age of the remains within the 19th century?
- Methods, Collagen Extraction: Include the following:
 - o Volume of 0.6 M HCl
 - o Volumes of 50 mM AmBic
 - o Mass/volume+concentration of trypsin added
 - o Brand of ultrafilters used
 - o Clarify the concentration and volumes of AmBic in the paragraph for archaeological ZooMS processing

o It would be useful for generalist readers to state in the relevant paragraphs that the methods are acid-insoluble and acid-soluble approaches respectively

- Section 2.4: Please give sufficient detail about the methods used for LC-MS/MS analysis. The facility should have some standardised text here. For example, how were the samples prepared for LC-MS/MS, what type of column was employed, and how was fractionation achieved?

- Page 5, line 22: “*Macropus sp.*” Should be “*Macropus spp.*”, if you searched against both the eastern and western grey kangaroos

- Results: At the beginning of the Results section, could you include a statement or paragraph that defines the level of identification possible from the collagen biomarkers for the taxa in the study. For example, I understand from Table 1 that not all species have a unique set of MALDI biomarkers (e.g. *Macropus fuliginosus* and *M. giganteus*; *Osphranter spp.* and others), but this doesn’t come clearly from the text on pages 6 and 7, nor from Table 1. Additionally, some taxa have evidence of confidently-assigned LC-MS/MS sequence markers that can help distinguish to species level. Am I right in thinking that generally speaking collagen can distinguish to genus-level in marsupials? The text is very detailed and sets out a large number of taxon-specific biomarkers (which is brilliant), but it’s currently hard to follow what the overall picture is with regards to the ability to refine species ID to a particular level (family, genus, species). I would also suggest that this should also be included as a sentence in the abstract. I now see that this statement begins the Discussion period. It would be useful to include information on this earlier in the article, in the abstract or results for example.

- Page 6, line 26 onwards: It could be interesting to include that the COL1A1 508-519 marker in many species of birds and reptile is also at m/z 1162 and compare the sequences. I believe Harvey et al., 2019 on Cayman Island fauna show this (citation below). The reason this may be necessary to include is due to the following sentence on page 8: “Seven samples were identified as marsupials on the basis of peptide marker COL1A1 508-519 (P1) but lacked other peptide markers to make more specific taxonomic identifications.” Could these be some form of reptile rather than marsupial? Were any reptiles other than turtle (which doesn’t have 1162) found in the morphological analyses?

- Page 6, lines 33 onwards: This is really interesting, and highlights potentially the only small issue with the new nomenclature system for peptide codes (which is otherwise extremely useful). For example, peptide COL1A2 454-483 (E) is presumably peptide COL1A2 454-487 (E) in monotremes, and the following peptide will have different numbers too, starting at A2 488. This is really common in fish, whereby peptides may be shortened significantly by R or K substitutions between different species, creating even more variation in m/z. Outlining which peptides are homologous using the new nomenclature is therefore slightly harder, but this is mostly overridden by the usefulness of having a set system in place for everyone to follow! This is just a note that you may wish to expand on in your manuscript.

- Page 7, line 23 onwards: These biomarkers, 1453 and 1427, are really interesting as they are the same (in mass at least) to many mammals. It could be good to include the sequence information in the Supplementary to show whether these sequences differ between macropods and other mammals? Or even in the text? This would be useful information to include to help researchers understand their own collagen fingerprints when a site contains both marsupials and placentals.

- Section 3.5, line 46 onwards: Please state how many samples have been identified as macropods in the text. Please state all the species that match the biomarkers observed. I.e. you have ruled some out and then suggested *L. conspicillatus* and *O. robustus isabellinus* based on prior morphological analyses done at the site, but it would be most applicable to explain to the reader all the species are still possible for these spectra and then lead on to suggest the most likely species. State where LC-MS/MS would be required for further refinement if this is the case (or include this in the Discussion).

- Page 8, line 6: You state that six specimens had spectra that were not attributable to any known taxa, and you suggest fish or birds. Do these samples appear in Table 4? I’m afraid it isn’t

currently clear which samples make up the 49 (37%) that generated viable collagen signatures. If you omit the birds/fish from this total then you either need to put them back in or be clearer in your opening sentence in section 3.5.

- Table 4 states 6 samples of ‘Marsupialia’ yet the text suggests there were 7 (line 7-8 on page 8).
- Page 8, lines 10 onwards: Include that you have been able to successfully identify bone fragments that were not able to be morphologically identified – one of the main strengths of the manuscript
- Discussion: I wouldn’t say the archaeological sample ID have particularly showcased the utility of the biomarkers particularly well. As I understand it, the identifications have largely relied on prior morphological analysis of the site remains, rather than the collagen signatures if they were to stand alone. Clearly the work to define the biomarkers will be hugely beneficial for future studies, and this is the real success of the work. But it does not do this with the archaeological samples in this study. I would therefore consider revising this sentence. Currently the archaeological samples do not indicate that there is ‘tremendous potential’ in using ZooMS on marsupial fauna. There probably will be great potential, but your samples do not demonstrate this.
- Section 4.3 “...reasonably well-preserved in the Bandicoot Bay assemblage...” – 37% of the assemblage produced a collagen signature, some of this was also poor quality, and the site is relatively modern. Unfortunately, I would say that the assemblage was not reasonably well preserved, but I also note that this statement is also subjective. Given the harsh environmental conditions, this level of poor preservation isn’t unsurprising, as you mention. You may wish to tone down the claim that it is reasonably well preserved but I will leave that to the author’s discretion.
- Discussion: Further comments on how evolutionally diverged members of the same genus or family are and how this corresponds to rate of sequence evolution would be welcomed.
- Table 3: Explain in the caption why some biomarkers are in bold
- Table 4: Please be clearer in the table caption that the morphological ID has come from prior analyses of the site and not of analyses of these particular bone fragments

References

Harvey, V.L., Egerton, V.M., Chamberlain, A.T., Manning, P.L., Sellers, W.I. and Buckley, M., 2019. Interpreting the historical terrestrial vertebrate biodiversity of Cayman Brac (Greater Antilles, Caribbean) through collagen fingerprinting. *The Holocene*, 29(4), pp.531-542.

Review form: Reviewer 2

Is the manuscript scientifically sound in its present form?

Yes

Are the interpretations and conclusions justified by the results?

Yes

Is the language acceptable?

Yes

Do you have any ethical concerns with this paper?

No

Have you any concerns about statistical analyses in this paper?

No

Recommendation?

Accept with minor revision (please list in comments)

Comments to the Author(s)

I have reviewed "RSOS-211229 Species Identification of Australian Marsupials using Collagen Fingerprinting". Building off earlier attempt to develop ZooMS for Australian megafauna, this paper presents a comprehensive list of collagen peptide markers that can be used to differentiate most marsupials from Australian archaeological contexts. The authors obtain collagen sequences from a broad range of modern taxa, and analyze these to determine a list of potential peptide markers. They then apply these to a faunal assemblage from an historical archaeological site on Barrow Island, Western Australia. Overall, I found the research to be well designed, with robust methods used to identify the peptide markers. I also appreciate how careful the authors were to note where multiple peptides can produce the same mass value, and how these may potentially confound ZooMS identifications. This paper really showcases the value of MS/MS data in identifying and validating ZooMS markers. The paper was clearly organized and well written. I have only minor recommendations to improve the clarity of some areas of the manuscript.

1) Methods: Archaeological specimens: The authors note the 'harsh' taphonomic conditions of the site preclude faunal identifications – I'm assuming this is due to high rates of fragmentation due to weathering and other processes. It would be useful to more clearly outline the taphonomic effects observed at the site, since this will also relate to collagen preservation in the assemblage on Barrow Island as well as other sites in Australia.

It would be useful to also analyze a subset of morphologically identified specimens in a blind test in order to validate both the ZooMS and morphological identifications from the previous analysis. Did you consider this as part of your experimental design?

2) Methods: Collagen extraction: Please add the volumes of HCl and Ambic used in modern and ancient collagen extractions.

3) Marsupials versus monotreme ZooMS markers: The authors note 3 peptide markers that may differentiate monotremes from marsupials. The authors only test one monotreme species however. Although it's likely that these peptides markers would be common among the family Tachyglossidae, I would be cautious about extrapolating these to Ornithorhynchidae. Considering that echidnas and platypuses may have split 19–48 million years ago it may be possible that these monotreme markers may differ in Ornithorhynchus. It would be worth noting here, or within section 4.3, the need to expand ZooMS to include other extant monotremes.

4) Overall, the authors are consistent in using Latin binomials in their discussion of taxonomic identifications. For those reader less familiar with these genus and species names, it can be useful to also provide common names. For example, in the discussion of the fauna identified at the archaeological site, including common names would help the reader more easily decipher the ZooMS findings with reference to the zooarchaeological data.

With reference to the bovid/cervid identified at the site – could you provide more detail on this identification? Which markers/potential species could this represent?

With reference to the potential fish or birds identified in the assemblage, were these spectra compared to potential bird peaks identified in Buckley et al. 2009 Rapid Commun. Mass

Spectrom, or fish peaks presented in Harvey et al. 2018, J. Arch. Sci.? Considering how few fish and bird ZooMS studies there are, readers would benefit from knowing whether these spectra contained any previously published Aves or conserved fish peaks.

5) Section 4.3 would benefit by specifically noting the need to expand to Australian carnivores, particularly the extensive family Dasyuridae. Considering the broad debates concerning human occupation and megafaunal extinction in the late Pleistocene, I would recommend highlighting how ZooMS might contribute to these debates, especially if extinct megafauna could also be identified through collagen fingerprints.

6) Finally, I applaud the authors for making their MS/MS and MALDI spectra publicly available. This is essential to advance the field, ensure replicability, and resolve inconsistencies in peptide markers between studies. This should be the standard for all ZooMS studies.

Minor suggestions:

Pg 3, Line 33: I would change 'alongside blanks' to 'alongside extraction blank controls'.

Pg 7, Section 3.4: It would be useful to remind the reader which taxa are included in 'macropods'

Table 2: It would be useful to provide diagnostic markers in bold font, particularly those markers that differentiate monotremes, or macropods.

Decision letter (RSOS-211229.R0)

Dear Miss Peters

On behalf of the Editors, we are pleased to inform you that your Manuscript RSOS-211229 "Species Identification of Australian Marsupials using Collagen Fingerprinting" has been accepted for publication in Royal Society Open Science subject to minor revision in accordance with the referees' reports. Please find the referees' comments along with any feedback from the Editors below my signature.

Please submit your revised manuscript and required files (see below) no later than 7 days from today's (ie 20-Sep-2021) date. Note: the ScholarOne system will 'lock' if submission of the revision is attempted 7 or more days after the deadline. If you do not think you will be able to meet this deadline please contact the editorial office immediately.

Please note article processing charges apply to papers accepted for publication in Royal Society Open Science (<https://royalsocietypublishing.org/rsos/charges>). Charges will also apply to papers transferred to the journal from other Royal Society Publishing journals, as well as papers

submitted as part of our collaboration with the Royal Society of Chemistry (<https://royalsocietypublishing.org/rsos/chemistry>). Fee waivers are available but must be requested when you submit your revision (<https://royalsocietypublishing.org/rsos/waivers>).

on behalf of Dr Michelle Alexander (Associate Editor) and Kevin Padian (Subject Editor)
openscience@royalsociety.org

Reviewer comments to Author:

Reviewer: 1

Comments to the Author(s)

Review of Peters et al., "Species Identification of Australian Marsupials using Collagen Fingerprinting"

Peters et al. present an extremely thorough and nicely written manuscript that sets out a suite of collagen type 1 biomarkers for 24 marsupials and a monotreme from Australia, the vast majority of biomarkers of which are new to the field. They then apply these biomarkers to a set of archaeological samples from Barrow Island (WA), from a 19th century pearlshell factory, in order to attempt collagen-derived taxonomic identification. The article vastly improves knowledge of marsupial (and monotreme) collagen biomarkers, as well as correcting some previously incorrectly assigned biomarkers from earlier in the 'evolution' of the technique. Overall, this is an extremely useful study that should be published immediately to give the greatest benefits to the zooarchaeological community. May I also thank the authors for depositing the raw MALDI and MS/MS data in an open access folder. I have no major issues with the manuscript, which was an enjoyable read, and just a few minor comments that I hope will strengthen the paper and be relatively easy for the authors to address.

Minor points

- Abstract: "We show that the application of ZooMS has the potential to significantly amplify the zooarchaeological and paleontological record of Australia" - I don't agree that your results for the archaeological assemblage in this study shows the potential to significantly amplify the zooarchaeological and paleontological record of Australia. I believe your biomarkers can, but your application of them in this case does not 'significantly amplify' the records of Australia. Your samples, with support of prior morphological analysis, increased the number of taxa known on the site from 4 to 6. This is not a large enough sample size to claim such significance. Please adapt this sentence to show the real strengths of the paper, which is the huge suite of biomarkers, rather than the specific application to the Barrow Island samples
- It would be useful for the readers if you were able to very briefly define 'macropods' in the Introduction
- Page 3, line 2: suggestion to include anthropogenic activities. I.e. "...enable reconstruction of palaeoenvironmental conditions and shifts in biodiversity over time, and help assess the impact of past climate change and anthropogenic activities."
- Page 3, line 34: You mention here that you are building on extant species, but you include extinct also, including thylacine. Would it be better to state "extant and extinct"?

- Page 4, lines 23 onwards: it is striking here that one of the species identified from the morphological assessments of the bone assemblage were not selected for the collagen reference collection. It would be most advantageous to acquire a modern bone sample of golden bandicoot (*I. auratus barrowensis*), but if this is outside of the scope of the current study (which I assume it is) then text should be included somewhere that explicitly notes and explains this omission. For example, in section 3.5.
- Methods: Can you include more precise information about the age of the remains within the 19th century?
- Methods, Collagen Extraction: Include the following:
 - o Volume of 0.6 M HCl
 - o Volumes of 50 mM AmBic
 - o Mass/volume+concentration of trypsin added
 - o Brand of ultrafilters used
 - o Clarify the concentration and volumes of AmBic in the paragraph for archaeological ZooMS processing
 - o It would be useful for generalist readers to state in the relevant paragraphs that the methods are acid-insoluble and acid-soluble approaches respectively
- Section 2.4: Please give sufficient detail about the methods used for LC-MS/MS analysis. The facility should have some standardised text here. For example, how were the samples prepared for LC-MS/MS, what type of column was employed, and how was fractionation achieved?
- Page 5, line 22: “*Macropus* sp.” Should be “*Macropus* spp.”, if you searched against both the eastern and western grey kangaroos
- Results: At the beginning of the Results section, could you include a statement or paragraph that defines the level of identification possible from the collagen biomarkers for the taxa in the study. For example, I understand from Table 1 that not all species have a unique set of MALDI biomarkers (e.g. *Macropus fuliginosus* and *M. giganteus*; *Osphranter* spp. and others), but this doesn’t come clearly from the text on pages 6 and 7, nor from Table 1. Additionally, some taxa have evidence of confidently-assigned LC-MS/MS sequence markers that can help distinguish to species level. Am I right in thinking that generally speaking collagen can distinguish to genus-level in marsupials? The text is very detailed and sets out a large number of taxon-specific biomarkers (which is brilliant), but it’s currently hard to follow what the overall picture is with regards to the ability to refine species ID to a particular level (family, genus, species). I would also suggest that this should also be included as a sentence in the abstract. I now see that this statement begins the Discussion period. It would be useful to include information on this earlier in the article, in the abstract or results for example.
- Page 6, line 26 onwards: It could be interesting to include that the COL1A1 508-519 marker in many species of birds and reptile is also at m/z 1162 and compare the sequences. I believe Harvey et al., 2019 on Cayman Island fauna show this (citation below). The reason this may be necessary to include is due to the following sentence on page 8: “Seven samples were identified as marsupials on the basis of peptide marker COL1A1 508-519 (P1) but lacked other peptide markers to make more specific taxonomic identifications.” Could these be some form of reptile rather than marsupial? Were any reptiles other than turtle (which doesn’t have 1162) found in the morphological analyses?
- Page 6, lines 33 onwards: This is really interesting, and highlights potentially the only small issue with the new nomenclature system for peptide codes (which is otherwise extremely useful). For example, peptide COL1A2 454-483 (E) is presumably peptide COL1A2 454-487 (E) in monotremes, and the following peptide will have different numbers too, starting at A2 488. This is really common in fish, whereby peptides may be shortened significantly by R or K substitutions between different species, creating even more variation in m/z. Outlining which peptides are homologous using the new nomenclature is therefore slightly harder, but this is mostly overridden by the usefulness of having a set system in place for everyone to follow! This is just a note that you may wish to expand on in your manuscript.

- Page 7, line 23 onwards: These biomarkers, 1453 and 1427, are really interesting as they are the same (in mass at least) to many mammals. It could be good to include the sequence information in the Supplementary to show whether these sequences differ between macropods and other mammals? Or even in the text? This would be useful information to include to help researchers understand their own collagen fingerprints when a site contains both marsupials and placentals.
- Section 3.5, line 46 onwards: Please state how many samples have been identified as macropods in the text. Please state all the species that match the biomarkers observed. I.e. you have ruled some out and then suggested *L. conspicillatus* and *O. robustus isabellinus* based on prior morphological analyses done at the site, but it would be most applicable to explain to the reader all the species are still possible for these spectra and then lead on to suggest the most likely species. State where LC-MS/MS would be required for further refinement if this is the case (or include this in the Discussion).
- Page 8, line 6: You state that six specimens had spectra that were not attributable to any known taxa, and you suggest fish or birds. Do these samples appear in Table 4? I'm afraid it isn't currently clear which samples make up the 49 (37%) that generated viable collagen signatures. If you omit the birds/fish from this total then you either need to put them back in or be clearer in your opening sentence in section 3.5.
- Table 4 states 6 samples of 'Marsupialia' yet the text suggests there were 7 (line 7-8 on page 8).
- Page 8, lines 10 onwards: Include that you have been able to successfully identify bone fragments that were not able to be morphologically identified – one of the main strengths of the manuscript
- Discussion: I wouldn't say the archaeological sample ID have particularly showcased the utility of the biomarkers particularly well. As I understand it, the identifications have largely relied on prior morphological analysis of the site remains, rather than the collagen signatures if they were to stand alone. Clearly the work to define the biomarkers will be hugely beneficial for future studies, and this is the real success of the work. But it does not do this with the archaeological samples in this study. I would therefore consider revising this sentence. Currently the archaeological samples do not indicate that there is 'tremendous potential' in using ZooMS on marsupial fauna. There probably will be great potential, but your samples do not demonstrate this.
- Section 4.3 "...reasonably well-preserved in the Bandicoot Bay assemblage..." – 37% of the assemblage produced a collagen signature, some of this was also poor quality, and the site is relatively modern. Unfortunately, I would say that the assemblage was not reasonably well preserved, but I also note that this statement is also subjective. Given the harsh environmental conditions, this level of poor preservation isn't unsurprising, as you mention. You may wish to tone down the claim that it is reasonably well preserved but I will leave that to the author's discretion.
- Discussion: Further comments on how evolutionally diverged members of the same genus or family are and how this corresponds to rate of sequence evolution would be welcomed.
- Table 3: Explain in the caption why some biomarkers are in bold
- Table 4: Please be clearer in the table caption that the morphological ID has come from prior analyses of the site and not of analyses of these particular bone fragments

References

Harvey, V.L., Egerton, V.M., Chamberlain, A.T., Manning, P.L., Sellers, W.I. and Buckley, M., 2019. Interpreting the historical terrestrial vertebrate biodiversity of Cayman Brac (Greater Antilles, Caribbean) through collagen fingerprinting. *The Holocene*, 29(4), pp.531-542.

Reviewer: 2

Comments to the Author(s)

I have reviewed "RSOS-211229 Species Identification of Australian Marsupials using Collagen Fingerprinting". Building off earlier attempt to develop ZooMS for Australian megafauna, this

paper presents a comprehensive list of collagen peptide markers that can be used to differentiate most marsupials from Australian archaeological contexts. The authors obtain collagen sequences from a broad range of modern taxa, and analyze these to determine a list of potential peptide markers. They then apply these to a faunal assemblage from an historical archaeological site on Barrow Island, Western Australia. Overall, I found the research to be well designed, with robust methods used to identify the peptide markers. I also appreciate how careful the authors were to note where multiple peptides can produce the same mass value, and how these may potentially confound ZooMS identifications. This paper really showcases the value of MS/MS data in identifying and validating ZooMS markers. The paper was clearly organized and well written. I have only minor recommendations to improve the clarity of some areas of the manuscript.

1) Methods: Archaeological specimens: The authors note the 'harsh' taphonomic conditions of the site preclude faunal identifications - I'm assuming this is due to high rates of fragmentation due to weathering and other processes. It would be useful to more clearly outline the taphonomic effects observed at the site, since this will also relate to collagen preservation in the assemblage on Barrow Island as well as other sites in Australia.

It would be useful to also analyze a subset of morphologically identified specimens in a blind test in order to validate both the ZooMS and morphological identifications from the previous analysis. Did you consider this as part of your experimental design?

2) Methods: Collagen extraction: Please add the volumes of HCl and Ambic used in modern and ancient collagen extractions.

3) Marsupials versus monotreme ZooMS markers: The authors note 3 peptide markers that may differentiate monotremes from marsupials. The authors only test one monotreme species however. Although it's likely that these peptide markers would be common among the family Tachyglossidae, I would be cautious about extrapolating these to Ornithorhynchidae. Considering that echidnas and platypuses may have split 19-48 million years ago it may be possible that these monotreme markers may differ in Ornithorhynchus. It would be worth noting here, or within section 4.3, the need to expand ZooMS to include other extant monotremes.

4) Overall, the authors are consistent in using Latin binomials in their discussion of taxonomic identifications. For those reader less familiar with these genus and species names, it can be useful to also provide common names. For example, in the discussion of the fauna identified at the archaeological site, including common names would help the reader more easily decipher the ZooMS findings with reference to the zooarchaeological data.

With reference to the bovid/cervid identified at the site - could you provide more detail on this identification? Which markers/potential species could this represent?

With reference to the potential fish or birds identified in the assemblage, were these spectra compared to potential bird peaks identified in Buckley et al. 2009 Rapid Commun. Mass Spectrom, or fish peaks presented in Harvey et al. 2018, J. Arch. Sci.? Considering how few fish and bird ZooMS studies there are, readers would benefit from knowing whether these spectra contained any previously published Aves or conserved fish peaks.

5) Section 4.3 would benefit by specifically noting the need to expand to Australian carnivores, particularly the extensive family Dasyuridae. Considering the broad debates concerning human occupation and megafaunal extinction in the late Pleistocene, I would recommend highlighting how ZooMS might contribute to these debates, especially if extinct megafauna could also be identified through collagen fingerprints.

6) Finally, I applaud the authors for making their MS/MS and MALDI spectra publicly available. This is essential to advance the field, ensure replicability, and resolve inconsistencies in peptide markers between studies. This should be the standard for all ZooMS studies.

Minor suggestions:

Pg 3, Line 33: I would change 'alongside blanks' to 'alongside extraction blank controls'.

Pg 7, Section 3.4: It would be useful to remind the reader which taxa are included in 'macropods'

Table 2: It would be useful to provide diagnostic markers in bold font, particularly those markers that differentiate monotremes, or macropods.

===PREPARING YOUR MANUSCRIPT===

===PREPARING YOUR REVISION IN SCHOLARONE===

Author's Response to Decision Letter for (RSOS-211229.R0)

See Appendix A.

Decision letter (RSOS-211229.R1)

Dear Miss Peters,

I am pleased to inform you that your manuscript entitled "Species Identification of Australian Marsupials using Collagen Fingerprinting" is now accepted for publication in Royal Society Open Science.

on behalf of Dr Michelle Alexander (Associate Editor) and Kevin Padian (Subject Editor)
openscience@royalsociety.org

Appendix A

Max-Planck-Institut für Menschheitsgeschichte

Max Planck Institute for the Science of Human History

MAX-PLANCK-GESellschaft

Carli Peters
peters@shh.mpg.de
+49 (0) 3641 686-965
MPI for the Science of Human History
Kahlaische Str. 10 • 07745 Jena

Dear Dr. Alexander,

We are pleased to resubmit our paper “Species Identification of Australian Marsupials using Collagen Fingerprinting” to *Royal Society Open Science*. We would like to thank you for your assistance and both reviewers for their time, effort and valuable comments and suggestions. We have carefully addressed the recommendations of the reviewers and believe the paper is stronger thanks to their helpful suggestions.

We provide a point by point response to specific reviewers’ comments below. We hope that our reply is clear and that this new version of the manuscript will be suitable for publication.

Yours sincerely,

Carli Peters

Rebuttal

Revisions of the main text are in blue

- Comments from Reviewer 1 –

Peters et al. present an extremely thorough and nicely written manuscript that sets out a suite of collagen type 1 biomarkers for 24 marsupials and a monotreme from Australia, the vast majority of biomarkers of which are new to the field. They then apply these biomarkers to a set of archaeological samples from Barrow Island (WA), from a 19th century pearlshell factory, in order to attempt collagen-derived taxonomic identification. The article vastly improves knowledge of marsupial (and monotreme) collagen biomarkers, as well as correcting some previously incorrectly assigned biomarkers from earlier in the ‘evolution’ of the technique. Overall, this is an extremely useful study that should be published immediately to give the greatest benefits to the zooarchaeological community. May I also thank the authors for depositing the raw MALDI and MS/MS data in an open access folder. I have no major issues with the manuscript, which was an enjoyable read, and just a few minor comments that I hope will strengthen the paper and be relatively easy for the authors to address.

Minor points

- Abstract: “We show that the application of ZooMS has the potential to significantly amplify the zooarchaeological and paleontological record of Australia” – I don’t agree that your results for the archaeological assemblage in this study shows the potential to significantly amplify the zooarchaeological and paleontological record of Australia. I believe your biomarkers can, but your application of them in this case does not ‘significantly amplify’ the records of Australia. Your samples, with support of prior morphological analysis, increased the number of taxa known on the site from 4 to 6. This is not a large enough sample size to claim such significance. Please adapt this sentence to show the real strengths of the paper, which is the huge suite of biomarkers, rather than the specific application to the Barrow Island samples

Response: Thank you for pointing this out. We have changed the abstract to tone down this claim, and instead focus on the potential that the peptide biomarkers have for the study of zooarchaeological and palaeontological assemblages from Australia.

- It would be useful for the readers if you were able to very briefly define ‘macropods’ in the Introduction

Response: We have added in a very brief definition of macropods in the introduction: “Amongst the best-recognized of Australia’s fauna are its marsupials, including macropods such as kangaroos and wallabies (members of the suborder Macropodiformes, generally characterized by their long powerful hind legs and feet), as well as other taxa such as koalas and wombats.”

- Page 3, line 2: suggestion to include anthropogenic activities. I.e. “...enable reconstruction of palaeoenvironmental conditions and shifts in biodiversity over time, and help assess the impact of past climate change and anthropogenic activities.”

Response: Thank you for the suggestion. We have changed the sentence accordingly.

- Page 3, line 34: You mention here that you are building on extant species, but you include extinct also, including thylacine. Would it be better to state “extant and extinct”?

Response: We have changed the sentence to say “extant and recently extinct marsupial and monotreme species”.

- Page 4, lines 23 onwards: it is striking here that one of the species identified from the morphological assessments of the bone assemblage were not selected for the collagen reference collection. It would be most advantageous to acquire a modern bone sample of golden bandicoot (*I. auratus barrowensis*), but if this is outside of the scope of the current study (which I assume it is) then text should be included somewhere that explicitly notes and explains this omission. For example, in section 3.5.

Response: Thank you for pointing out this omission. We have added in an explicit statement on this in section 3.5 with the aim to explain why a modern bone sample of the golden bandicoot was not acquired for this study.

- Methods: Can you include more precise information about the age of the remains within the 19th century?

Response: We have added in a more detailed age estimate of the site, which is estimated to have been occupied somewhere in the 1880s/1890s (see Dooley et al 2020).

- Methods, Collagen Extraction: Include the following:
 - o Volume of 0.6 M HCl
 - o Volumes of 50 mM AmBic
 - o Mass/volume+concentration of trypsin added
 - o Brand of ultrafilters used
 - o Clarify the concentration and volumes of AmBic in the paragraph for archaeological ZooMS processing
 - o It would be useful for generalist readers to state in the relevant paragraphs that the methods are acid-insoluble and acid-soluble approaches respectively

Response: We changed section 2.2. to include these details about the method.

- Section 2.4: Please give sufficient detail about the methods used for LC-MS/MS analysis. The facility should have some standardised text here. For example, how were the samples prepared for LC-MS/MS, what type of column was employed, and how was fractionation achieved?

Response: We have expanded section 2.4. to give more detailed about the methods used for LC-MS/MS analysis.

- Page 5, line 22: “*Macropus* sp.” Should be “*Macropus* spp.”, if you searched against both the eastern and western grey kangaroos

Response: We did not search against both the eastern and the western grey kangaroo. Instead, the data we searched against was not identified to species level. Therefore, we will retain “*Macropus* sp.” as it is in text.

- Results: At the beginning of the Results section, could you include a statement or paragraph that defines the level of identification possible from the collagen biomarkers for the taxa in the study. For example, I understand from Table 1 that not all species have a unique set of MALDI biomarkers (e.g. *Macropus fuliginosus* and *M. giganteus*; *Osphranter* spp. and others), but this doesn't come clearly

from the text on pages 6 and 7, nor from Table 1. Additionally, some taxa have evidence of confidently-assigned LC-MS/MS sequence markers that can help distinguish to species level. Am I right in thinking that generally speaking collagen can distinguish to genus-level in marsupials? The text is very detailed and sets out a large number of taxon-specific biomarkers (which is brilliant), but it's currently hard to follow what the overall picture is with regards to the ability to refine species ID to a particular level (family, genus, species). I would also suggest that this should also be included as a sentence in the abstract. I now see that this statement begins the Discussion period. It would be useful to include information on this earlier in the article, in the abstract or results for example.

Response: Thank you for this suggestion. We have added a sentence in the abstract and at the beginning of the results section that defines the general level of identification possible for the taxa in the study.

- Page 6, line 26 onwards: It could be interesting to include that the COL1A1 508-519 marker in many species of birds and reptile is also at m/z 1162 and compare the sequences. I believe Harvey et al., 2019 on Cayman Island fauna show this (citation below). The reason this may be necessary to include is due to the following sentence on page 8: “Seven samples were identified as marsupials on the basis of peptide marker COL1A1 508-519 (P1) but lacked other peptide markers to make more specific taxonomic identifications.” Could these be some form of reptile rather than marsupial? Were any reptiles other than turtle (which doesn't have 1162) found in the morphological analyses?

Response: Thank you for pointing this out. We have added a sentence to include the m/z value of the COL1A1 508-519 marker in birds and reptiles. We compared the sequences, which turn out to be identical. We have also mentioned this in the revised text.

With regards to the faunal identifications at Bandicoot Bay, turtle was the only reptile identified morphologically from the historical assemblage. There are a number of terrestrial reptiles on the island, however. Following the comparison of peptide marker COL1A1 508-519, we changed this section to “Seven samples only showed a peak at peptide marker COL1A1 508-519 (P1) at m/z 1162, but lacked other peptide markers to make taxonomic identifications”. Accordingly, we have also deleted these identifications from Table 4 and changed the number of identified specimens in the text throughout.

- Page 6, lines 33 onwards: This is really interesting, and highlights potentially the only small issue with the new nomenclature system for peptide codes (which is otherwise extremely useful). For example, peptide COL1A2 454-483 (E) is presumably peptide COL1A2 454-487 (E) in monotremes, and the following peptide will have different numbers too, starting at A2 488. This is really common in fish, whereby peptides may be shortened significantly by R or K substitutions between different species, creating even more variation in m/z. Outlining which peptides are homologous using the new nomenclature is therefore slightly harder, but this is mostly overridden by the usefulness of having a set system in place for everyone to follow! This is just a note that you may wish to expand on in your manuscript.

Response: This is definitely worth noting and something to keep in mind. We have added a few sentences to the manuscript to elaborate on this comment.

- Page 7, line 23 onwards: These biomarkers, 1453 and 1427, are really interesting as they are the same (in mass at least) to many mammals. It could be good to include the sequence information in the Supplementary to show whether these sequences differ between macropods and other mammals? Or even in the text? This would be useful information to include to help researchers understand their own collagen fingerprints when a site contains both marsupials and placentals.

Response: We have added the sequence information to the text in section 3.4. Furthermore, the sequence information for these peptide markers can also be found in Table 2.

- Section 3.5, line 46 onwards: Please state how many samples have been identified as macropods in the text. Please state all the species that match the biomarkers observed. I.e. you have ruled some out and then suggested *L. conspicillatus* and *O. robustus isabellinus* based on prior morphological analyses done at the site, but it would be most applicable to explain to the reader all the species are still possible for these spectra and then lead on to suggest the most likely species. State where LC-MS/MS would be required for further refinement if this is the case (or include this in the Discussion).

Response: Thank you for pointing out this lack of clarity in our line of reasoning. We have added in the number of samples that has been identified as macropods. We also added in a list of all macropod species that could be possible for the spectra.

- Page 8, line 6: You state that six specimens had spectra that were not attributable to any known taxa, and you suggest fish or birds. Do these samples appear in Table 4? I'm afraid it isn't currently clear which samples make up the 49 (37%) that generated viable collagen signatures. If you omit the birds/fish from this total then you either need to put them back in or be clearer in your opening sentence in section 3.5.

Response: Thanks for pointing this out. These samples do not appear in Table 4, since no reliable taxonomic identification was possible for these samples. We have added in a sentence in section 3.5 to more clearly describe that these samples are not included in the table, or in the % of samples with sufficient collagen preservation to make taxonomic identifications.

- Table 4 states 6 samples of 'Marsupialia' yet the text suggests there were 7 (line 7-8 on page 8).

Response: In line with the earlier comment about the COL1A1 508-519 peptide marker, we have removed the identification to Marsupialia altogether (from both the text and Table 4). This resolves this issue as well.

- Page 8, lines 10 onwards: Include that you have been able to successfully identify bone fragments that were not able to be morphologically identified – one of the main strengths of the manuscript

Response: Thank you for the suggestion, we have added this in.

- Discussion: I wouldn't say the archaeological sample ID have particularly showcased the utility of the biomarkers particularly well. As I understand it, the identifications have largely relied on prior morphological analysis of the site remains, rather than the Wcollagen signatures if they were to stand alone. Clearly the work to define the biomarkers will be hugely beneficial for future studies, and this is the real success of the work. But it does not do this with the archaeological samples in this study. I would therefore consider revising this sentence. Currently the archaeological samples do not indicate that there is 'tremendous potential' in using ZooMS on marsupial fauna. There probably will be great potential, but your samples do not demonstrate this.

Response: Thank you for this suggestion. We have changed the sentence to focus less on the results of the case study, and more on the promise that the peptide biomarkers hold for future studies.

- Section 4.3 "...reasonably well-preserved in the Bandicoot Bay assemblage..." – 37% of the

assemblage produced a collagen signature, some of this was also poor quality, and the site is relatively modern. Unfortunately, I would say that the assemblage was not reasonably well preserved, but I also note that this statement is also subjective. Given the harsh environmental conditions, this level of poor preservation isn't unsurprising, as you mention. You may wish to tone down the claim that it is reasonably well preserved but I will leave that to the author's discretion.

Response: Thank you for this comment. We have changed the sentence to better reflect the preservation conditions at Bandicoot Bay "Collagen preservation in the Bandicoot Bay assemblage was variable; in some specimens collagen was reasonably well-preserved and spectra were suitable to make taxonomic identifications, while poorer preservation in other specimens meant that resolution of spectra was not sufficient for ZooMS identifications."

- Discussion: Further comments on how evolutionally diverged members of the same genus or family are and how this corresponds to rate of sequence evolution would be welcomed.

Response: Thank you for this suggestion. We have added in a short paragraph in which we examine the evolutionary divergence of the different macropod genera and how this corresponds to differences we observed in the collagen sequences of these genera.

- Table 3: Explain in the caption why some biomarkers are in bold

Response: Thank you for pointing out this inconsistency. The table caption has been changed to explain why some peptide markers are bolded.

- Table 4: Please be clearer in the table caption that the morphological ID has come from prior analyses of the site and not of analyses of these particular bone fragments

Response: Thank you for pointing out our lack of clarity here. We changed the sentence to make it clearer.

- Comments from Reviewer 2 -

I have reviewed "RSOS-211229 Species Identification of Australian Marsupials using Collagen Fingerprinting". Building off earlier attempt to develop ZooMS for Australian megafauna, this paper presents a comprehensive list of collagen peptide markers that can be used to differentiate most marsupials from Australian archaeological contexts. The authors obtain collagen sequences from a broad range of modern taxa, and analyze these to determine a list of potential peptide markers. They then apply these to a faunal assemblage from an historical archaeological site on Barrow Island, Western Australia. Overall, I found the research to be well designed, with robust methods used to identify the peptide markers. I also appreciate how careful the authors were to note where multiple peptides can produce the same mass value, and how these may potentially confound ZooMS identifications. This paper really showcases the value of MS/MS data in identifying and validating ZooMS markers. The paper was clearly organized and well written. I have only minor recommendations to improve the clarity of some areas of the manuscript.

1) Methods: Archaeological specimens: The authors note the 'harsh' taphonomic conditions of the site preclude faunal identifications – I'm assuming this is due to high rates of fragmentation due to weathering and other processes. It would be useful to more clearly outline the taphonomic effects observed at the site, since this will also relate to collagen preservation in the assemblage on Barrow Island as well as other sites in Australia.

Response: Thank you for the suggestion. We have added more information about the taphonomic effects observed at Bandicoot Bay, mostly relating to bone fragmentation rates.

It would be useful to also analyze a subset of morphologically identified specimens in a blind test in order to validate both the ZooMS and morphological identifications from the previous analysis. Did you consider this as part of your experimental design?

Response: This is something that would definitely be useful. We decided not to do this because we did not want to destroy the limited sample of identifiable bone that is available for the site.

2) Methods: Collagen extraction: Please add the volumes of HCl and Ambic used in modern and ancient collagen extractions.

Response: We have added in the volumes of HCl and Ambic used for the extractions, as well as some further details about the methods used for the collagen extraction of modern and archaeological specimens.

3) Marsupials versus monotreme ZooMS markers: The authors note 3 peptide markers that may differentiate monotremes from marsupials. The authors only test one monotreme species however. Although it's likely that these peptides markers would be common among the family Tachyglossidae, I would be cautious about extrapolating these to Ornithorhynchidae. Considering that echidnas and platypuses may have split 19–48 million years ago it may be possible that these monotreme markers may differ in Ornithorhynchus. It would be worth noting here, or within section 4.3, the need to expand ZooMS to include other extant monotremes.

Response: Thank you for pointing this out. We have added a sentence in section 3.2 to state that one should be cautious when extrapolating these results to Ornithorhynchidae. In section 4.3 we have included a sentence highlighting the need to develop ZooMS peptide markers for other extant monotremes.

4) Overall, the authors are consistent in using Latin binomials in their discussion of taxonomic identifications. For those reader less familiar with these genus and species names, it can be useful to also provide common names. For example, in the discussion of the fauna identified at the archaeological site, including common names would help the reader more easily decipher the ZooMS findings with reference to the zooarchaeological data.

Response: Thank you for this suggestion. We have added the common names of species when discussing the results of ZooMS analysis at Bandicoot Bay (sections 3.5 and 4.1) to make this discussion more accessible for readers.

With reference to the bovid/cervid identified at the site – could you provide more detail on this identification? Which markers/potential species could this represent?

Response: Thank you for this question. We have added in a sentence in section 3.5 to provide more detail about the possible identification of this specimen.

With reference to the potential fish or birds identified in the assemblage, were these spectra compared to potential bird peaks identified in Buckley et al. 2009 Rapid Commun. Mass Spectrom, or fish peaks

presented in Harvey et al. 2018, J. Arch. Sci.? Considering how few fish and bird ZooMS studies there are, readers would benefit from knowing whether these spectra contained any previously published Aves or conserved fish peaks.

Response: We did compare the spectra to published bird and fish peaks, but there do not appear to be any distinctive bird or fish peaks that could help to resolve this issue.

5) Section 4.3 would benefit by specifically noting the need to expand to Australian carnivores, particularly the extensive family Dasyuridae. Considering the broad debates concerning human occupation and megafaunal extinction in the late Pleistocene, I would recommend highlighting how ZooMS might contribute to these debates, especially if extinct megafauna could also be identified through collagen fingerprints.

Response: Thank you for this suggestion. We have added in a couple of sentences highlighting how ZooMS might be able to contribute to the debate on megafaunal extinctions in Australia.

6) Finally, I applaud the authors for making their MS/MS and MALDI spectra publicly available. This is essential to advance the field, ensure replicability, and resolve inconsistencies in peptide markers between studies. This should be the standard for all ZooMS studies.

Minor suggestions:

Pg 3, Line 33: I would change ‘alongside blanks’ to ‘alongside extraction blank controls’.

Response: We have changed this sentence accordingly.

Pg 7, Section 3.4: It would be useful to remind the reader which taxa are included in ‘macropods’

Response: We have briefly reiterated which taxa are classed as macropods.

Table 2: It would be useful to provide diagnostic markers in bold font, particularly those markers that differentiate monotremes, or macropods.

Response: We have changed the table accordingly. Bolded masses in the table can be used to differentiate between taxa.